



# Measurement report: Introduction to the HyICE-2018 campaign for measurements of ice nucleating particles in the Hyytiälä boreal forest.

Zoé Brasseur[1], Dimitri Castarède[2], Erik S. Thomson[2], Michael P. Adams[3], Saskia Drossaart van Dusseldorp[4,**], Paavo Heikkilä[5], Kimmo Korhonen[6], Janne Lampilahti[1], Mikhail Paramonov[4], Julia Schneider[7], Franziska Vogel[7], Yusheng Wu[1], Jonathan P. D. Abbatt[8], Nina S. Atanasova[9,10], Dennis H. Bamford[10], Barbara Bertozzi[7], Matthew Boyer[1], David Brus[9], Martin I. Daily[3], Romy Fösig[7], Ellen Gute[8], Alexander D. Harrison[3], Paula Hietala[1], Kristina Höhler[7], Zamin A. Kanji[4], Jorma Keskinen[5], Larissa Lacher[7], Markus Lampimäki[1], Janne Levula[1], Antti Manninen[9], Jens Nadolny[7], Maija Peltola[1], Grace C. E. Porter[3,11], Pyry Poutanen[1], Ulrike Proske[3,12,*], Tobias Schorr[7], Nsikanabasi Silas Umo[7], János Stensky[2], Annele Virtanen[6], Dmitri Moisseev[1,9], Markku Kulmala[1], Benjamin J. Murray[3], Tuukka Petäjä[1], Ottmar Möhler[7], and Jonathan Duplissy[1,13]

[1]Institute for Atmospheric and Earth System Research/Physics, Faculty of Science, University of Helsinki, Helsinki, Finland
[2]Department of Chemistry and Molecular Biology, Atmospheric Science, University of Gothenburg, Gothenburg, Sweden
[3]Institute for Climate and Atmospheric Science, School of Earth and Environment, University of Leeds, Leeds, UK
[4]Institute for Atmospheric and Climate Science, ETH Zurich, Switzerland
[5]Aerosol Physics Laboratory, Physics Unit, Faculty of Engineering and Natural Sciences, Tampere University, Tampere, Finland
[6]Department of Applied Physics, University of Eastern Finland, Kuopio, Finland
[7]Institute of Meteorology and Climate Research, Karlsruhe Institute of Technology, Karlsruhe, Germany
[8]Department of Chemistry, University of Toronto, Toronto, Ontario, Canada
[9]Finnish Meteorological Institute, Helsinki, Finland
[10]Molecular and Integrative Biosciences Research Programme, Faculty of Biological and Environmental Sciences, University of Helsinki, Finland
[11]School of Physics and Astronomy, University of Leeds, Leeds, UK
[12]Institute for Atmospheric and Environmental Sciences, Goethe University Frankfurt, Frankfurt am Main, Germany
[13]Helsinki Institute of Physics, University of Helsinki, Helsinki, Finland
[*]now at: Institute for Atmospheric and Climate Science, ETH Zurich, Switzerland
[**]now at: Centre for Aviation, Zurich University of Applied Sciences, Winterthur, Switzerland

**Correspondence:** Erik S. Thomson (erik.thomson@chem.gu.se), Jonathan Duplissy (jonathan.duplissy@helsinki.fi) and Zoé Brasseur (zoe.brasseur@helsinki.fi)

**Abstract.** The formation of ice particles in Earth's atmosphere strongly influences the dynamics and optical properties of clouds and their impacts on the climate system. Ice formation in clouds is often triggered heterogeneously by ice nucleating particles (INPs) that represent a very low number of particles in the atmosphere. To date, many sources of INPs, such as mineral and soil dust, have been investigated and identified in the lower latitudes. Although less is known about the sources of ice nucleation at higher latitudes, efforts have been made to identify the sources of INPs in the Arctic and boreal environments. In this study, we investigate the INP emission potential from high latitude boreal forests. We introduce the HyICE-2018 measurement campaign conducted in the boreal forest of Hyytiälä, Finland between February and June 2018. The campaign utilized the infrastructure





of the SMEAR II research station with additional instrumentation for measuring INPs to quantify the concentrations and sources of INPs in the boreal environment. In this contribution, we describe the measurement infrastructure and operating procedures during HyICE-2018 and we report results from specific time periods where INP instruments were run in parallel for inter-comparison purposes. Our results show that the suite of instruments deployed during HyICE-2018 reports consistent

results and therefore lays the foundation for forthcoming results to be considered holistically. In addition, we compare the INP concentration we measured to INP parameterizations, and we show a very good agreement with the Tobo et al. (2013) parameterization developed from measurements conducted in a ponderosa pine forest ecosystem in Colorado, USA.

## 1   Introduction

Atmospheric aerosols are recognized to play an important role in nearly every aspect of the physics and chemistry of the

atmosphere (Solomon et al., 2007; Boucher et al., 2013). However, the interactions between aerosols and clouds and how these interactions influence Earth's surface energy budget and water cycle represent significant knowledge gaps in climate science. Thus, it is critical to understand fundamental aerosol processes such as aerosol formation, growth and aerosol-cloud interactions to evaluate the impact of aerosols on Earth's radiative balance.

Certain types of atmospheric particles, called ice nucleating particles (INPs), have the potential to initiate the formation of

ice in clouds, thus affecting the properties of the clouds and often the initiation of precipitation. Heterogeneous ice nucleation processes may include (i) deposition nucleation where bulk liquid water is presumed to be absent and ice is formed from vapor supersaturated with respect to ice; (ii) immersion freezing where ice formation is initiated by an INP located within a body of liquid; (iii) condensation freezing where an INP simultaneously acts as a cloud condensation nucleus (CCN); and (iv) contact freezing where ice formation is triggered at the air-water interface by an INP that comes into contact with a supercooled liquid

droplet (Vali et al., 2015).

Numerous measurements of INPs have been performed in various settings and environments. However, measurements in boreal forests are largely underrepresented and little is known concerning the INP sources and properties from this environment. Boreal forests represent more than one-third of all forests and cover more than 15 million km$^2$ of land (Tunved et al., 2006). The majority of boreal landscape is in the Arctic and sub-Arctic region of the continental Northern Hemisphere, and

due to its high-latitude position, the environment experiences strong seasonal changes in meteorological and environmental conditions. Moreover, the boreal forest environment is generally far from large dust sources and strong anthropogenic emissions, which motivates the investigation of biogenic ice nucleation activity in this environment. Finally, INPs from mid to high latitude sources may have a disproportionate effect on climate through their influence on shallow clouds (Murray et al., 2021), emphasizing the need to know more about INP sources in these regions, including in boreal forests.

To overcome our lack of knowledge, a measurement campaign was organized in 2018 at the Station for Measuring Ecosystem-Atmosphere Relations (SMEAR) II, at the Hyytiälä Forestry Research Station in Juupajoki, Finland. The intensive field campaign, called HyICE-2018, began in February and extended until June, with longer term INP monitoring efforts continuing for more than one year beyond the intensive measurement period (Schneider et al., 2021).





One aim of the campaign was to utilize the highly instrumented and time-resolved measurements of the SMEAR II station. The station is a well known atmospheric observatory in a high latitude location (61°51'N) within a boreal forest ecosystem (Hari and Kulmala, 2005). The SMEAR II site has hosted field campaigns for decades and is well suited for investigating potential links between atmospheric and ecosystem processes. The station is equipped with a suite of advanced aerosol instru-

mentation that has been utilized for significant advancements in aerosol process studies. Therefore, the SMEAR II station is well suited for an intensive campaign focused on measuring INPs in the boreal forest environment.

Additionally, new particle formation (NPF) events are frequently recorded at the SMEAR II station (Kulmala et al., 2013), and there are relevant scientific questions that remain unanswered concerning the role of NPF in the context of ambient INPs. NPF events occur when volatile organic compounds (VOCs) and biogenic volatile organic compounds (BVOCs) are oxidized

and condensed to form very small particulate, secondary organic aerosol (SOA), some of which continue to grow to larger sizes that may form cloud condensation nuclei (CCN) and INPs (Kulmala et al., 2013). The role of NPF in forming particles large enough to participate in cloud activation as CCN and INPs has been investigated in laboratory studies (Duplissy et al., 2008; Möhler et al., 2008; Ladino et al., 2014; Ignatius et al., 2016). However, while there is evidence that NPF can play a role in CCN activation (Frosch et al., 2011; Sihto et al., 2011), and studies have shown that organic aerosol can nucleate ice (Murray

et al., 2010; Wilson et al., 2012; Wagner et al., 2017; Wolf et al., 2020), the relationship between NPF and INPs has not been explored with field measurements.

In order to augment the standard SMEAR II monitoring instrumentation for the HyICE-2018 campaign, several institutions deployed INP measurement systems, including the University of Helsinki, the University of Gothenburg, the Karlsruhe Institute of Technology, the University of Leeds, the University of Eastern Finland and the Swiss Federal Institute of Technology in

Zürich (ETH Zürich). Furthermore, the Finnish Meteorological Institute and Tampere University contributed remote sensing retrievals and bioaerosol monitoring, respectively. In addition to the individual data collection efforts, several days were utilized for instrument inter-comparison analyses to test the reproducibility of results across instruments and scientific teams in the field setting.

The primary objectives of the campaign were to

• quantify and characterize INPs in a boreal environment within different thermodynamic forcing regimes (i.e., different temperatures, $T$, and supersaturations with respect to ice, $S_i$)

• determine and quantify the existence of any seasonal variation of INPs and

• assess the vertical distribution of INPs above the boreal forest.

This paper gives an overview of the campaign setting and design as an introduction to the Copernicus Special Issue, "Ice

nucleation in the boreal atmosphere", which is anticipated to include several contributions from HyICE-2018. Data from several days dedicated to evaluate instrument inter-comparison is presented to illustrate instrument-to-instrument agreement and to facilitate future presentation and interpretation of data from subsets of instruments.



## 2  Methods

### 2.1  Measurement Site – SMEAR II

The HyICE-2018 campaign took place at the SMEAR II station in Hyytiälä, Finland (Hari and Kulmala, 2005). The station is located in Southern Finland (61°51'N, 24°17'E; 181 m a.s.l.) and is surrounded by boreal coniferous forest (Fig. 1). The conditions at the site are typical for a background location, with the main pollution sources being the city of Tampere (60 km to the southwest with $\approx 238,000$ inhabitants as of 2019) and the activity and buildings at the station (Kulmala et al., 2001; Boy et al., 2004). The station has several operational units that span a wide forest area and reach into and above the tree canopy. Multiple towers include a 128 m mast used for atmospheric flux measurements, a 18 m tower for irradiation and flux measurements, a separate 18 m tower for tree physiology measurements and a 35 m walk-up tower for aerosol measurements. There are also several measurement cottages and containers spread throughout the forest, with a cottage located near the walk-up tower and dedicated to measuring the physical properties of aerosols, and a main cottage based near the 128 m mast (Fig. 1).

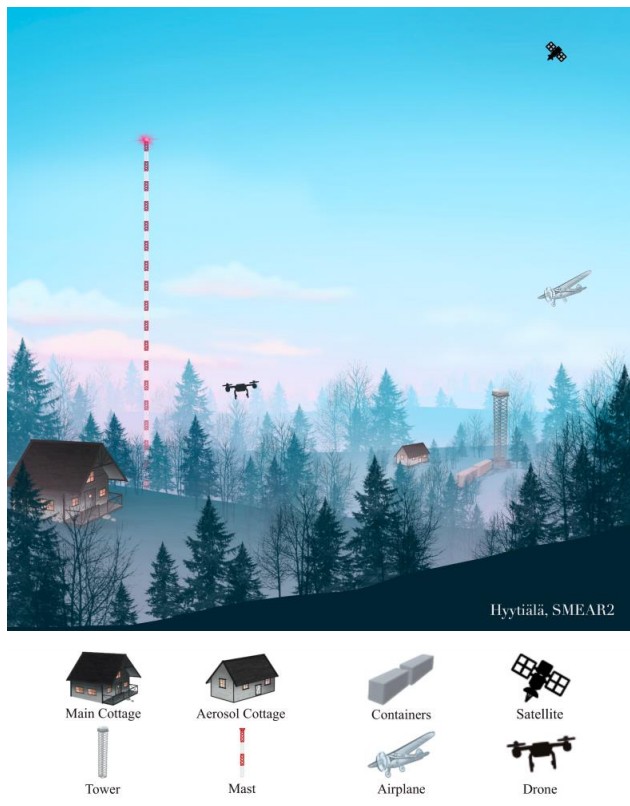

**Figure 1.** Overview of the SMEAR II station in Hyytiälä, Finland used for staging the HyICE-2018 measurement campaign, with the available infrastructures and measurements platforms depicted.





**Table 1.** Instrumentation installed at the SMEAR II research station specifically for the HyICE-2018 campaign. A list of the instruments permanently running at the SMEAR II research station can be found in Tables A1, A2 and A3.

| Function | Type | Instrument | Institution | Inlet | Time Resolution | Sampling location |
|---|---|---|---|---|---|---|
| INP counters | Continuous Flow Diffusion Chamber (CFDC) | PINC | ETH | $PM_{2.5}$+ heated inlet + dryer | 20-30 min NC | Main cottage |
| | | PINCii | INAR, GU | $PM_{2.5}$+ heated inlet | 15 min NC | Main cottage |
| | | SPIN (DMT) | UEF | $PM_{2.5}$+ dryer | 30 min NC | Container |
| | Expansion chamber | PINE | KIT, Leeds | Heated inlet | 6 min continuous | Main cottage |
| | Droplet Freezing Assay | INSEKT | KIT | $PM_{10}$ | 10-24 hrs | Aerosol cottage, 35 m tower, onboard aircraft |
| | | μL-NIPI | Leeds | $PM_1$, $PM_{2.5}$, $PM_{10}$ | 4-12 hrs | |
| Particle counters | Condensation Particle Counter | CPC 3010 (TSI) | INAR | TSP | continuous | 128 m mast |
| | | CPC 3010 (TSI) | INAR | TSP | continuous | 35 m tower |
| | Optical Particle Sizer | OPS 3330 (TSI) | INAR | TSP | continuous | 128 m mast |
| | Bioaerosol sensor | WIBS-NEO (DMT) | TAU | $\geq 0.5\,\mu m$ | continuous | cottages |

NC = Not Continous; continuous = measuring without interruption unless maintenance was required.

The SMEAR II station is equipped to monitor the physical and chemical properties of aerosols and gas phase precursors to aerosol formation with a suite of state-of-the-art monitoring instrumentation. Measurements also cover meteorology, radiation, soil, snow cover and gases. An overview of the instruments in operation at the site during the HyICE-2018 campaign is available in the supplemental materials (Tables A1,A2 and A3). In this study, data from the SMEAR II Differential Mobility

Particle Sizer (DMPS) and Aerosol Particle Sizer (APS; TSI model 3321) were used. The DMPS measures aerosol particle size distributions from 3 to 1000 nm in mobility diameter, with a 10 min time resolution (Aalto et al., 2001; Jokinen and Mäkelä, 1997). During HyICE-2018, the instrument was sampling through a total suspended particle inlet 8 m inside the forest canopy and was operated following the guidelines from Aerosols, Clouds and Trace gases Research InfraStructure (ACTRIS; Wiedensohler et al. 2012). The APS is used to measure the super-micron aerosol particle size distribution from 0.5 to 20 μm in

aerodynamic diameter. The instrument was sampling through a total suspended particle inlet (DIGITEL Elektronik GmbH) 6 m above ground level, and a vertical sampling line was used to avoid particle losses. In addition, the APS inlet was heated to 40 °C to ensure that the relative humidity in the sampling inlet remained below 40%, which prevents condensation and dries the aerosols before measurements.

  The instruments deployed specifically for HyICE-2018 are summarized in Table 1, and the dates they were operational are

depicted in Fig. 2 against a background of the changing seasons illustrated by the evolving average daytime temperatures $\overline{T}_{\mathrm{day}}$ (mean surface temperature 08.00–20.00, UTC+2). The campaign-specific instrumentation combined with the measurements from SMEAR II provides a rich dataset for the analysis of INPs. Additionally, the large parameter space allows for advanced machine learning techniques to be applied, where the parameter space dimensionality is reduced to illuminate processes strongly linked to INP signals, even where connections are not obvious. Such advanced analyses are explored in the Wu

et al. (2021) manuscript in preparation, among others.

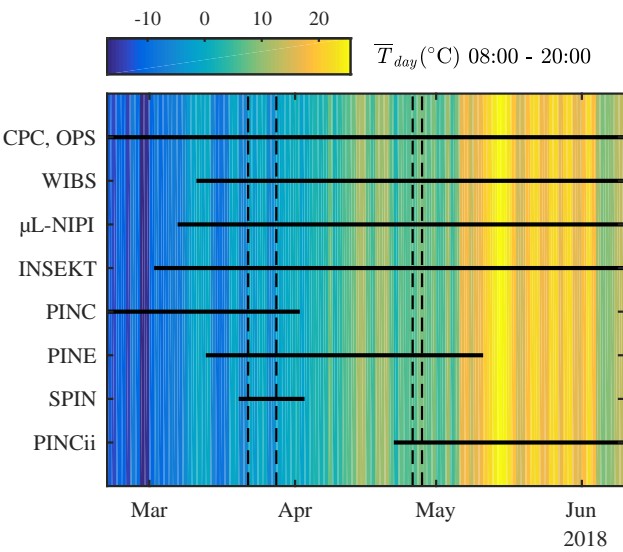

**Figure 2.** Timeline of deployment for the instrumentation installed specifically for the HyICE-2018 campaign. Shading depicts mean daytime air temperature $\overline{T}_{day}$ (08:00 – 20:00, UTC+2) measured at 4.2 m height. Instrument inter-comparison days are indicated by the dashed lines.

## 2.2 Augmented sampling for Ice Nucleating Particles

One of the motivations behind HyICE-2018 was to compare different INP measurement techniques in a field setting. Previous studies have performed inter-comparisons of INP instrumentation in a number of intensive measurement campaigns (DeMott et al., 2011; Wex et al., 2015; Hiranuma et al., 2015; Burkert-Kohn et al., 2017; DeMott et al., 2018; Hiranuma et al., 2019).

However, these efforts focused on well-controlled laboratory measurements to assess sampling procedures and to calibrate instruments relative to one another (Hiranuma et al., 2015; Wex et al., 2015). Only a few INP instruments have been co-located for long field measurements of real atmospheric aerosol (DeMott et al., 2017), and continued efforts in multi-instrument measurements can be beneficial for the entire community (Lacher et al., 2020).

During HyICE-2018, several instruments and techniques were used to quantify INPs. The working principles of the different

INP instruments are depicted in Fig. 3. Three Continuous Flow Diffusion Chambers (CFDCs, described in §2.2.1) and one expansion chamber (described in §2.2.2) were used to conduct online measurements of INP concentrations, and two droplet freezing assays (described in §2.2.3) were used for offline analysis of INPs collected on filters. The intensive measurement campaign took place between February and June 2018, and each instrument's collection period is illustrated in Fig. 2.

### 2.2.1 Continuous Flow Diffusion Chambers

Three CFDCs with parallel-plate designs (PINC, PINCii, SPIN) were deployed for online INP measurements (Fig. 3a). The three models are iterations of a design that consists of two parallel ice-coated walls which are cooled to below freezing temperatures (Rogers, 1988; Stetzer et al., 2008; Chou et al., 2011). In the upper part of the chamber, referred to as the main chamber,





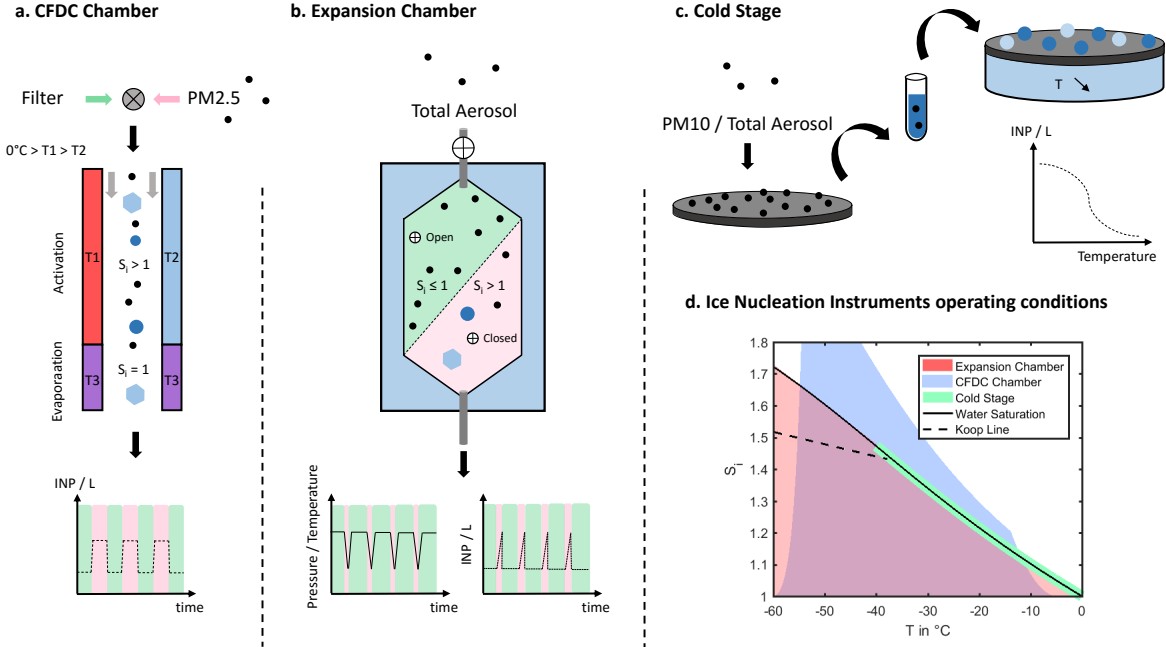

**Figure 3.** Schematic of the working principles of the INP measurement systems deployed during HyICE-2018. (a) The CFDC chambers operate with activation and evaporation sections, and alternate ambient sampling (pink) and filtered background measurements (green). (b) The PINE expansion chamber samples a volume of ambient air before a short expansion cycle activates INP within the trapped volume. In both (a) and (b), optical particle counting is used at the chamber exits to measure particle number. (c) The droplet freezing assays rely on filter sampling of ambient aerosols. Collected particles are washed from the filters and analyzed for INP content, generating curves of INP temperature spectra. (d) The thermodynamic space typically accessible to different measurement techniques.

the walls are held at different temperatures (Fig. 3a), to produce vapor supersaturation in the region between the walls (Rogers, 1988). The lower chamber, or evaporation section, is held at an isothermal condition of ice saturation. During measurements, a continuous flow of sample air, referred to as the sample lamina, is sandwiched between two particle-free sheath flows and is drawn through the center of the chamber, thus exposing any airborne particles to the supersaturated conditions. At a given

5  lamina temperature, $T_l$, and saturation condition (e.g., supersaturation with respect to ice, $S_i$) some particles will induce water condensation and/or ice formation by accommodating excess vapor. When the flow leaves the main supersaturated chamber, it immediately enters an evaporation section, which is held at ice saturation, and is thus sub-saturated with respect to liquid water. Within the evaporation section, liquid droplets evaporate, creating a size difference between frozen and unfrozen particles. When paired with an inlet size cutoff, particles within the exit flow that exceed the cutoff are determined to be ice. Above

10  a certain $S_i$, CFDCs have an instrument-specific point of "droplet breakthrough" where the residence time of the evaporation section does not enable adequate evaporation for phase differentiation based on cut-off size with standard optical particle





counting. During HyICE-2018, the CFDCs were operated at relative humidities below the point of droplet breakthrough in order to prevent such a situation.

CFDCs are online instruments that measure INP concentration in real time with a minimum time resolution determined by the instrument specific particle counting method. However, since ambient INP concentrations are generally low, measurements typically consist of multi-minute counting averages. For the CFDCs used in this study, sampling intervals of 5 to 20 minutes were separated by background measurements of clean, filtered air. Overall, CFDC measurements are time-limited by the quality of the thin ice layer coating the chamber walls, which deteriorates over time and contributes to increasing particle counts as the instruments operate. Single experiments typically last three to five hours, after which the ice coating needs to be regenerated to restore low background conditions. This is done by warming and purging the chamber, before re-cooling and re-coating the walls with ice. This process can last one to three hours, allowing for two to four daily measurement cycles if continuous operation is desired (Paramonov et al., 2020). Concentrations obtained during background measurements are subtracted from the concentrations measured during each sampling window to compute the measured INP concentrations, where the lower limit of detection is defined as one standard error above the background mean. In the following sections we describe the instrument specifics for each CFDC run during HyICE-2018.

**I. Portable Ice Nucleation Chamber – PINC**

The Portable Ice Nucleation Chamber (PINC) was the first generation field-deployable, parallel-plate CFDC based on the designs of Stetzer et al. (2008), and it has been operated in many locations around the world for more than a decade (Chou et al., 2011, 2013; Boose et al., 2016; Kanji et al., 2019). During HyICE-2018, PINC was operated from February 19 to April 2 (Fig. 2) in the main aerosol cottage (Figs. 1 & 4), with one to two experimental cycles conducted per day, always during daytime. For the duration of the campaign PINC was operated at a fixed lamina temperature of $T_l$ = -31 °C and a relative humidity with respect to water $RH_w = 105\%$. These conditions were selected to simulate mixed-phase cloud conditions and correspond to the condensation/immersion freezing mode(s) of ice nucleation (Vali et al., 2015). Sampling was performed from a total aerosol inlet mounted outside of the building, 6 m above ground level (Fig. 4). The inlet was heated to 25-30 °C to evaporate droplets and ice crystals and sampled ambient air with a flow rate of 250 L min$^{-1}$. From the large inlet flow, a smaller 4 L min$^{-1}$ sample flow was extracted, and a cyclone was used to eliminate particles larger than 2.5 μm. A molecular sieve dryer was installed to further reduce the sample relative humidity. After the dryer, the sample flow was split into four: 1 L min$^{-1}$ to a Condensation Particle Counter (CPC; TSI model 3010), 1 L min$^{-1}$ to an Aerodynamic Particle Sizer (APS; TSI model 3321), 1 L min$^{-1}$ to a Scanning Mobility Particle Sizer (SMPS; with one Hauke-type differential mobility analyzer (DMA) and one TSI model 3772 CPC) and 1 L min$^{-1}$ to PINC (Paramonov et al., 2020).

On several occasions, a Portable Fine Particle Concentrator (PFPC) was used to concentrate aerosol particles in the sample flow upstream of PINC (Fig. 4). The PFPC, described by Gute et al. (2019) and based on the design by Sioutas et al. (1995), is a multi-stage concentrator based on virtual impaction. It concentrates aerosol particles with a certain size-dependent enrichment factor where larger particles are concentrated more efficiently than smaller ones (Gute et al., 2019). For HyICE-2018, the size-dependent enrichment factor was determined by measuring the particle size distributions before and after the PFPC, as





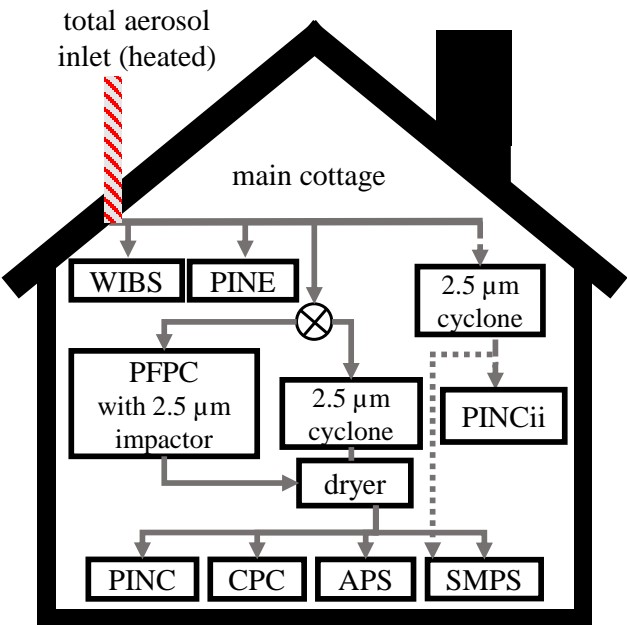

**Figure 4.** Instrumental setup in the main cottage. Note that PINC and its complete setup were used from February 19 to April 2, 2018 and were replaced by PINCii from April 22 to June 10, 2018 (see text for details).

explained in Paramonov et al. (2020). Ambient INP concentrations were back-calculated by multiplying the concentrated INP concentrations by a second enrichment factor determined before each experiment.

When the PFPC was used, the 2.5 μm cyclone was removed and replaced by the 2.5 μm impactor located inside the PFPC (see Fig. 4). On March 22, for inter-comparison purposes, the cyclone was removed so that PINC directly sampled the inlet air, and the operating conditions were changed to $T_l$ = -29 °C and $RH_w = 105\%$.

In this article we summarize the performance of PINC during the targeted days of instrument inter-comparison, while the longer-term results from the PINC measurements during HyICE-2018 are presented and discussed by Paramonov et al. (2020).

**II. Portable Ice Nucleation Chamber II – PINCii**

The Portable Ice Nucleation Chamber II (PINCii) is a parallel-plate CFDC developed as an upgrade to the PINC instrument. Although many specific engineering details differ, the primary differences between PINC and PINCii are the chamber dimensions and cooling power. While PINC's main chamber and evaporation section are 568 and 230 mm in height respectively (Chou et al., 2011), PINCii is approximately twice as large with a main chamber of 1000 mm and an evaporation section of 440 mm. A manuscript outlining the engineering and experimental/operational details of PINCii is in preparation (Castarède et al., 2021).

During the campaign, PINCii was operational from April 22 to June 10, 2018 and measured INP concentrations at a fixed $T_l$=-32 °C and $RH_w = 105\%$ in order to generate results comparable with the earlier PINC measurements. PINCii was located in the main cottage and sampled from the heated total aerosol inlet, essentially acting as a substitute for the early PINC





measurements as depicted in Fig. 4. However, during this later sampling interval, the PFPC was not used and the additional SMPS was used for the first two weeks of June only. For the typical setup during the campaign, PINCii sampled downstream of the 2.5 μm cyclone.

### III. Spectrometer for Ice Nuclei – SPIN

The Spectrometer for Ice Nuclei (SPIN, DMT) is a commercially developed CFDC based on the parallel-plate PINC design. The design and use of SPIN has been previously documented (Garimella et al., 2016, 2017, 2018) and some SPIN instruments ($\approx$ 10 exist worldwide) have participated in earlier inter-comparison activities (DeMott et al., 2018). For HyICE-2018, the University of Eastern Finland SPIN (UEF-SPIN) instrument was installed and operated for several days between March 20 and April 3. SPIN was situated in a measurement container approximately 200 m from the main cottage (see Fig. 1). SPIN

sampled through a burled $PM_{2.5}$ inlet (Digitel Enviro-sense Inc., Switzerland) located 110 cm above the roof of the container. A virtual impactor (VI)–type concentrator was used to enhance SPIN's sampling and detection of INPs. The $PM_{2.5}$ inlet had a nominal sampling flow rate of 16.67 L min$^{-1}$ regulated by an external pump and mass-flow controllers (MFCs) (Fig. 5). The configuration illustrated in Fig. 5 provided an amplification of the particle number concentration by approximately nine-fold. A CPC (Airmodus model A20) was operated in parallel with the SPIN to monitor the concentrated particle number concentration

during measurements.

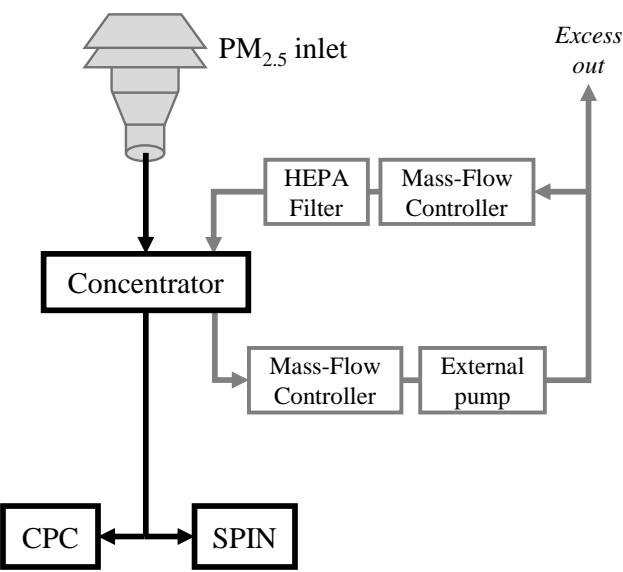

**Figure 5.** SPIN instrumental setup during the HyICE-2018 campaign.

The use of the VI-concentrator may have generated biases in the SPIN measurements. Indeed, the magnification factor of the VI-concentrator was most prominent for particles with diameters from 1.3 to 2.5 μm, with an average of 8.45±0.43. For smaller particles, the magnification factor decreased steeply towards 1, being 4.2 for 850 nm particles. This resulted in larger





super-micron particles being over-represented in the sampled particles in comparison to their ambient number concentrations. On average, the APS data showed very low number concentrations for particle diameters larger than 1.3 µm, and it can be expected that the role of such particles as INPs is more pronounced in the SPIN observations. Note that, due to the absence of correction method for the SPIN measurements during HyICE-2018, the SPIN data presented here is the concentrated INP

concentration per volume of sampled air and not the back-calculated ambient INP concentration.

During its first three measurement days, SPIN was programmed to measure using automated RH-scans where the relative humidity with respect to water, $RH_w$, was increased from 70 to 110% while keeping a constant lamina temperature. Such scans were realized for lamina temperatures of $T_l$= -40 °C, -36 °C and -32 °C. On these three days, a total of 33 RH-scans were performed with 5-minute background checks performed between successive scans. Thereafter, the sampling program was

modified to be more suitable for ambient particle measurements, and a modified scanning protocol was used for the remainder of the campaign. The modified sampling protocol consisted of longer scans at $T_l$ = -32 °C and -28 °C, where at each $T_l$, measurements were made with two saturation conditions ($RH_w$= 95% and 110%). At each of the 4 conditions, 20-minute sampling intervals followed by 5 minutes of background sampling were used. Measurements continued until the background signal exceeded 10-15 particles $L^{-1}$, after which the chamber was re-iced to return to a lower background signal (Garimella

et al., 2016, 2017, 2018).

For inter-comparison purposes, SPIN sampled at $T_l$ = -29 °C and $RH_w$ = 105% on March 22, and $T_l$ = -31 °C and $RH_w$ = 105% on March 28.

### 2.2.2  Expansion Chamber – PINE

The Portable Ice Nucleation Experiment (PINE) chamber (Fig. 3b), as described in (Möhler et al., 2021), has been developed

based on the working principle of the Aerosol Interactions and Dynamics in the Atmosphere (AIDA) cloud chamber (Bunz et al., 1996; Möhler et al., 2003, 2006), which simulates cloud formation in rising atmospheric air parcels. The PINE chamber is operated in a cycled mode during which it is first flushed with ambient aerosol particles to renew the sampled volume of air under investigation. During HyICE-2018, this first mode was run for 4 minutes with a flow rate of 3 L min$^{-1}$. Then, the expansion mode is initiated by sealing the main inlet valve and pumping air out of the chamber, leading to a decreasing pressure

within the chamber. As the pressure decreases to 700 mb in approximately 40 seconds, the volume of gas expands and the gas temperature decreases, leading to an increase in the relative humidity within the chamber. Upon reaching water saturation, aerosol particles within the chamber activate to form supercooled liquid droplets. If aerosol particles immersed within the droplets are also active as INPs at the respective droplet temperature, then those droplets will freeze via the immersion freezing pathway. Frozen droplets are optically detected at the exit of the chamber using an Optical Particle Counter (OPC, Welas model

2500p) and are differentiated from liquid droplets based on their size. After the expansion, a final refill mode is conducted, wherein the chamber continues to be depressurized using dry, filtered air to avoid any icing. The total experimental time of the 3-mode cycle is approximately 6 minutes and each cycle generates one measurement point. Longer averaging times (e.g., 1/6/24 hr) are commonly implemented in post-processing for purposes of statistical analysis. PINE is operated in a way that ensures frost-free walls such that a subtraction of any background ice counts is not needed.



During HyICE-2018, PINE was located in the main cottage (Figs. 1 and 4) and was operated continuously from March 13 until May 11, 2018 at measurement temperatures between -24 °C and -32 °C. During this time, PINE was operating with either a constant temperature or with a stepwise temperature ramping, during which the temperature was lowered three times by 2 - 3 °C each hour. In the main cottage, PINE sampled from the heated total aerosol inlet without using a size cut-off in order to

5 sample the total aerosol. Particle transmission efficiency as a function of particle size (Fig. 6) was investigated by measuring particle concentrations at the inlet of the PINE chamber and in the ambient air using an OPC (MetOne model GT 526S). The results illustrate that PINE was effectively sampling $PM_5$ aerosol; although the cut-off was not as sharp as would be expected from a traditional impactor.

During the inter-comparison days, PINE was operating at a constant temperature close to the lamina temperature, $T_l$, selected

for the CFDCs. On March 22, PINE sampled at a temperature of $T$=-29 °C while on March 28, the temperature was lowered to $T$=-30 °C. For the inter-comparison conducted in April, PINE sampled at $T$=-29 °C.

While this study focuses on PINE's results during the inter-comparison days, the long-term measurements will be presented and discussed in the Adams et al. (2022) manuscript in preparation.

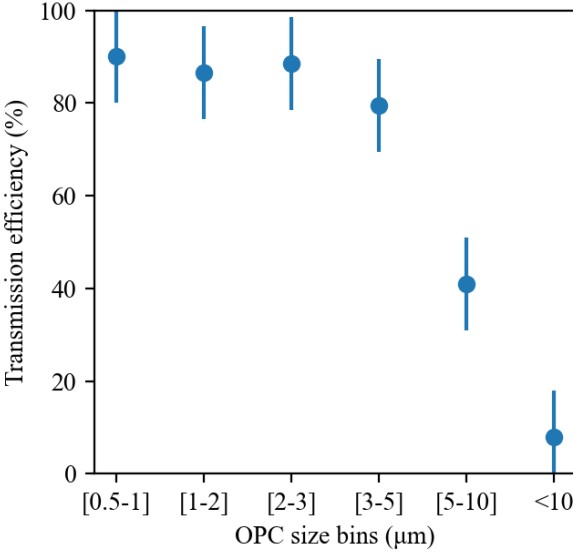

**Figure 6.** Transmission efficiency as a function of particle size for the PINE sampling system. The measurements were made with an OPC (MetOne, GT 526S) with an accuracy of $\pm 10\%$.

### 2.2.3 Filter sampling for droplet freezing assays

Numerous droplet freezing assay techniques exist for offline measurements of INPs in sampled aerosol. Techniques vary in terms of aerosol collection methodology, which can be filter collection (Conen et al., 2012), particle impaction into liquid (Šantl-Temkiv et al., 2017), electrostatic deposition (Schrod et al., 2016) and/or collection of bulk materials (Hill et al., 2014,



2016). All sampling techniques presented here rely on the re-suspension of collected material into known volumes of liquid (Fig. 3c). The liquid is generally separated into sample aliquots which are exposed to decreasing temperatures and monitored for freezing using optical techniques. Aliquots are assumed to freeze when the most "active" INP in any given sample volume initiates ice formation. A series of assumptions allows the concentration of INP at a given freezing temperature to be calculated

from the number of frozen aliquots (Vali, 1971b). It is important to note that these techniques typically assess the nucleation tendencies of sample material that is immersed within the liquid (Fig. 3d). However, the strength of droplet freezing assay techniques is that the sampling and analysis can be disconnected and thus relatively simple sampling units can be deployed with the labor intensive analysis done at dedicated laboratory facilities. Another advantage of droplet freezing assay techniques is that they provide a complete INP temperature spectra, as opposed to the INP concentration at one specific temperature.

During HyICE-2018, two droplet freezing assay techniques using similar sampling protocols but different freezing systems were deployed, and the analyses were performed in the SMEAR II laboratories.

**I. Ice Nucleation SpEctrometer of the Karlsruhe Institute of Technology – INSEKT**

The INSEKT is a droplet freezing assay based on the Colorado State University Ice Spectrometer (CSU-IS) design, which was originally developed by Hill et al. (2014, 2016). A detailed description of the INSEKT setup and working principle can be

found in Schiebel (2017). For INSEKT, filters were collected beginning March 2 and continuing through the end of the intensive campaign in mid-June 2018. Further INSEKT filters were collected more intermittently at the site for more than one more year (until May 2019), and that data has yielded detailed information related to the seasonality of observed INPs (Schneider et al., 2021). Ambient aerosol particles were collected onto 0.2 μm filters (Whatman nuclepore track-etched polycarbonate membranes, 47 mm), which were pre-cleaned with a 10% $H_2O_2$ solution and subsequently rinsed with deionized water that

was passed through a 0.1μm syringe filter (Whatman). After collection, the filters were stored in sterile petri dishes, wrapped in aluminum foil and frozen until analysis with INSEKT. Sampling was performed at three different SMEAR II locations. The primary sampling unit was installed at the aerosol cottage (Fig. 1) at ground level, where daily 24h filter samples were collected through a $PM_{10}$ inlet mounted on a ≈1.8 m vertical sampling line. A secondary filter sampling unit was used in a container on top of the 35 m tower, again using a $PM_{10}$ inlet and ≈1.8 m vertical sampling line. Generally, from the 35 m tower, two

filters were collected per day (≈10 hours during the day and ≈14 hours during the night). However, there were 10 days during the campaign when the tower sampling time was extended to 24 hours in order to directly compare with the results from the aerosol cottage. On March 28, two filter samples were collected at the main cottage for inter-comparison with the other INP instruments. For both filters, the sampling time was approximately 3 hours, and sampling was carried out using a total aerosol inlet.

After sampling, the collected aerosol particles were washed off the filters and suspended into 8 ml of nanopure water that had been passed through a 0.1 μm syringe filter. Small volumes of 50 μL were pipetted into the 192 wells of two PCR plates. The plates were then inserted in the freezing apparatus and were cooled with a constant 0.25 or 0.33 °C min⁻¹ cooling rate using an ethanol chiller (LAUDA Dr. R. Wobser GmbH & Co. KG.; Proline RP 890 in combination with a LAUDA command module). Brightness changes of the small sample volumes, which correspond to freezing events, were detected using a camera and an

image acquisition and analysis software. The number of frozen volumes, which increases as the temperature decreases, is used





to determine a temperature spectrum of INP concentration (Vali, 1971a). The INSEKT is able to measure INP concentrations at temperatures between -5 and -25 °C, which is relevant for heterogeneous freezing conditions within supercooled mixed-phase clouds.

After quantifying the INP content of the aerosol samples, some heat treatment tests were performed to investigate the heat-
sensitivity of the sampled aerosol with respect to their freezing ability, which can be used to investigate if biological particles contributed to the INP population (Hill et al., 2016). For the heat treatment, the liquid aerosol suspensions were placed in boiling water (100 °C) for about 20 minutes. The heat-treated samples were then re-analyzed with the INSEKT to quantify changes in the INP temperature spectra.

**II. microlitre Nucleation by Immersed Particle Instrument – μL-NIPI**

The μL-NIPI, similar to INSEKT, is also a droplet freezing assay instrument for offline measurements of INP concentrations requiring the collection of aerosol samples on filters. During HyICE-2018, filters were collected from March 7 until June 10, 2018 using omnidirectional ambient air particulate samplers (BGI PQ100, Mesa Laboratories Inc.). Filters were collected at ground level next to the aerosol cottage using $PM_{10}$, $PM_{2.5}$ and $PM_1$ inlets on 0.4 μm pore track-etched membrane polycarbonate filters (Nuclepore, Whatman). Filters were also collected using a prototype version of the Selective Height Aerosol
Research Kit (SHARK) (Porter et al., 2020) to quantify more detailed, size resolved measurements of the INP population, although this was done only sporadically for a few different sampling periods.

After sampling, collected aerosol particles were washed from filters using 5 mL of nanopure water that had been filtered through a 0.2 μm filter (Sartorius, model Minisart). Droplets of the sample solution containing particles were pipetted onto a hydrophobic glass slide that holds approximately 50 droplets of 1 μL (Fig. 3c). The glass slide was placed on the temperature-
controlled plate of the μL-NIPI, which was cooled to -40 °C at 1 °C per minute. The freezing temperature of each droplet was recorded via a digital camera using changes in contrast to determine when a droplet had frozen (Whale et al., 2015).

Heat tests were performed on the samples by increasing the aerosol suspension temperature to 100 °C for 30 minutes (Hill et al., 2016; O'Sullivan et al., 2018). The heated samples were analyzed using the μL-NIPI and the results were compared to the original unheated sample results in order to quantify any changes in INP activity, which is used to infer information about
INPs of biogenic origin.

## 2.3   Additional Aerosol Particle Characterization

Beyond INP measurements, further enhanced efforts were made to quantify, classify and assess aerosol properties at SMEAR II during the HyICE-2018 campaign. Multiple additional particle counters were installed to provide physical characterization of particles (Table 1). In addition to these ground and tower based instruments, airplane, drone flights and remote sensing
retrievals were used to provide insight into the vertical distribution of aerosols and INPs. Moreover, the SMEAR II station has a unique boreal location and is known for the documented occurrences of NPF events driven by biogenic volatile organic compounds emitted by the vegetation of the boreal forest (Lehtipalo et al., 2018). Hence, extra effort was made to assess whether any links between biology and INP emerge, with a special focus on the seasonal transition of the forest biome.





### 2.3.1 Search for bio-ice nucleators

The boreal forest is a diverse ecosystem in which biological drivers of aerosol properties have been previously identified. For example, past research at the SMEAR II research station has shown that gas phase BVOCs can act as precursors to NPF (Kulmala et al., 2013). In addition, previous studies have identified several biological influences on INPs, including biological detritus (Hiranuma et al., 2019; O'Sullivan et al., 2015), pollen (Dreischmeier et al., 2017), bacteria such as Pseudomonas Syringae (Morris et al., 2004), fungi (Morris et al., 2013), and other microorganisms such as viruses (Adams et al., 2021). Therefore, during HyICE-2018, we aimed to assess potential links between biological activity and INPs.

In order to evaluate INPs of biological origin, a Wideband Integrated Bioaerosol Sensor (WIBS; Droplet Measurement Technology model WIBS-NEO) was used for online analysis of particles during the campaign. The WIBS measures particles between 0.5 and 30 μm using a light scattering technique. In addition to particle counting, particles trigger two optically filtered (280 nm and 370 nm) Xenon lamps that excite the fluorescence of biological particles. The emission is monitored in two detection bands (310-400 nm and 420-650 nm), whereafter an additional fluorescence threshold can be used to distinguish biological particles (strong fluorescence response) from other materials (e.g., some types of dust, soot, black carbon) that are typically more weakly fluorescent. In this study, a threshold of $FT + 9\sigma$ (where FT is the mean value of the forced trigger intensities and $\sigma$ is their standard deviation) was used to determine if a particle is fluorescent (details in Savage et al. 2017). The use of two excitation wavelengths and two detection channels allows for additional resolution in the fluorescence analysis, because different particle types often have different fluorescence intensities and emission bands (Savage et al., 2017). An additional benefit of the WIBS instrument is that it allows particle asphericity to be calculated (Savage et al., 2017). During HyICE-2018, the WIBS was operated from March 11 to June 25 at a flow rate of 0.3 L min$^{-1}$ with an acquisition rate of 13 Hz, and the data was later averaged into user-defined time intervals of 10 minutes. The WIBS was first installed in the main cottage (March 11 – April 3; Fig. 4) where it was sampling from the total aerosol inlet, as previously described. Later, the instrument was moved into the aerosol cottage (April 3 – June 25) and attached to a PM$_{10}$ inlet described in Schmale et al. (2017).

In addition to online measurements, several offline sampling techniques were used to investigate potential biological contributions to INP from primary biological materials. Plant materials were collected and washed to test for ice nucleation active biological materials, and additional air samples were collected to cultivate fungal samples for DNA sequencing. For fungal cultivation, air samples were collected overnight from May 21 to May 22 and May 22 to May 23 on Teflon filters using a flow rate of 20 L min$^{-1}$. After sampling, the filters were suspended in a buffer solution (100 mM NaCl, 0.5 mM CaCl$_2$, 1.0 mM MgCl$_2$, 20 mM K-phosphate pH 7.2, filter sterilized through 0.22 μm pore size), and 200 μL of the sample were spread on Luria-Berthani plates and incubated at 22 °C for $\approx$ 7 days. Single filamentous colonies of different morphology were isolated and inoculated on fresh media and subsequently incubated at 22 °C. The ice nucleation activity of each filamentous colony was then tested using the μL-NIPI, and the results are presented in the Atanasova et al. (2021) manuscript in preparation.





### 2.3.2 Vertical Profiling

The vertical distribution of INP through the atmospheric mixed layer is a key area of investigation within the INP research community. During HyICE-2018, several techniques were used in order to assess if vertical gradients exist in INP concentrations, which might bias ground-level measurements.

Airborne measurements were made between March and May 2018 using a Cessna 172 aircraft equipped with on-board instrumentation for classifying aerosol physical properties and equipment for INP filter sampling for offline characterization using INSEKT and μL-NIPI. In-flight monitoring of aerosol physical properties was conducted using a Particle Size Magnifier (PSM; Airmodus model A10) and a CPC (TSI model 3776) to measure particle number concentrations. Furthermore, a SMPS and an Optical Particle Sizer (OPS; TSI model 3330) were used to classify particle number size distributions. A shrouded

solid diffuser inlet, designed based on the University of Hawaii inlet (McNaughton et al., 2007), with a 5.0 μm aerodynamic diameter cut-off was used. With this setup, particle number size distributions from 1.5 nm to 5.0 μm were measured. Additional sensors measured relative humidity, temperature and $CO_2$ and $H_2O$ concentrations (Li-Cor Li-840) during the flights. The aircraft's GPS receiver recorded latitude, longitude and flight altitude. In addition to characterizing the ambient conditions, these measurements were used to estimate the boundary layer depth. Filter samples were collected within the boundary layer

and the free troposphere to characterize the vertical distribution of INP concentrations. Measurement flights took off from the nearby ($\approx$ 60 km distant) Tampere-Pirkkala airport, with 3 hour-long flight plans that consisted of 20-40 km long segments flown over the SMEAR II research station at different altitudes between 100 and 3500 m a.g.l.

    Drone flights flown by the Finnish Meteorological Institute's - Remotely Piloted Aircraft System (FMI-RPAS) rotorcraft hexacopter were conducted on March 27 in an effort to assess the state of lower boundary layer aerosol and for comparisons

with the instruments installed at various heights on the measurement towers. The RPAS carried a payload that included two CPCs (TSI model 3007) with different cut-off diameters (7 nm and 14 nm), an OPC (Alphasense model N2, 16 size bins from 0.38 to 17 μm) to classify particles by size, and sensors to measure temperature, relative humidity (Vaisala HMP110 probe) and pressure (Arduino Bosch BME280). More details on the platform and the instrumentation setup are given in Brus et al. (2021). During HyICE-2018, flights were typically conducted below 700 m a.g.l.

The SMEAR II station is additionaly a cloud profiling station of ACTRIS equipped with a cloud radar (94 GHz FMCW Doppler Cloud Radar, RPG-FMCW-94-DP), a Doppler lidar (HALO Photonics DL) and a profiling microwave radiometer (Humidity And Temperature PROfiler, RPG-HATPRO). The cloud profiling instruments were operational during the HyICE-2018 campaign. The cloud observations are used to quantify cloud properties such as cloud boundaries, phase, ice water content and liquid water content (Illingworth et al., 2007). Moreover, radar observations can be used to identify ice particle growth

processes like riming (Kneifel and Moisseev, 2020) and the onset of ice particle formation (Oue et al., 2015; Li and Moisseev, 2020). In combination with INP measurements, these observations are useful for identifying cases that may be attributed to secondary ice production (Field et al., 2017; Sinclair et al., 2016).





## 3    Results and Discussion

The wide range of techniques and instrumentation employed during HyICE-2018 have led to many results. In this paper, we summarize the general results for the duration of the intensive campaign and focus on a few instances where defined efforts were made to inter-compare different INP measurement techniques and instrumentation. Longer term and more detailed studies that

have emerged from specific instruments and/or activities have been or will be published independently, with many contributions aimed at the Copernicus special issue, "Ice nucleation in the boreal atmosphere" (Paramonov et al., 2020; Schneider et al., 2021).

### 3.1    Meteorological Conditions and Seasonal Change

The HyICE-2018 campaign aimed to capture the seasonal transition from winter to summer conditions. As expected for the

boreal region, during winter the campaign was characterized by deep snow cover and cold temperatures (Fig. 7). In 2018, the transition from winter to summer was rather abrupt at the measurement site, and as depicted in Fig. 7, snow cover went from a near maximum to completely melted away within a couple of weeks in April 2018. This transition coincides with an increase in the fraction of fluorescent particles (Fig. 8), which are used as a proxy for biological particles (Savage et al., 2017). By mid-May, the forest ecosystem was fully transitioned into summer and ambient temperatures reached nearly 30 °C. In fact,

May 2018 was declared anomalously warm over all of Finland (Sinclair et al., 2019). The seasonal change is also noticeable from particle number size distribution measurements (Fig. 9), with an increase in particle concentrations beginning in April. The seasonal change is less evident in the NPF event frequency, which is also plotted in Fig. 9 (white diamonds; Dal Maso et al., 2005; Vana et al., 2008). Analysis of aerosol characteristics (concentration and NPF occurrence) as a function of wind direction, where trends might be expected due to varying source regions (Tunved et al., 2003, 2006), do not show a coupling

with season either (Fig. 10). Such observations further motivate the search for other seasonally dependent variables, such as aerosol chemical composition and biological activity, that may influence the ice nucleating potential of the aerosols (Tobo et al., 2013).

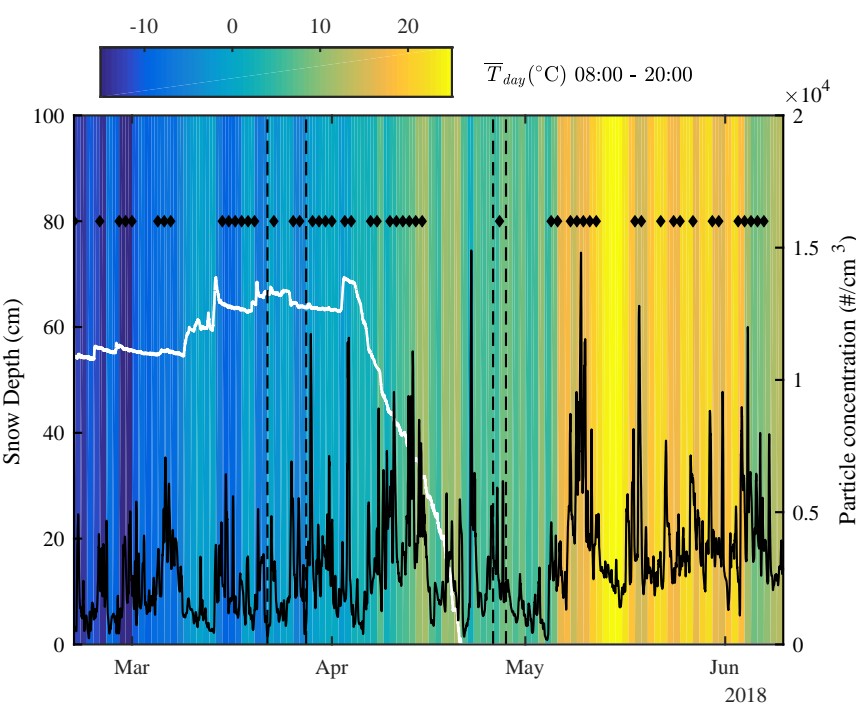

**Figure 7.** Snow depth (1 min resolution, white trace), particle concentration (8 hr moving average calculated from the SMEAR II DMPS and APS total concentrations, black line) and NPF events (black diamonds) as a function of time for the duration of the HyICE-2018 campaign. Shading depicts the mean daytime air temperature as already presented in Fig. 2. Instrument inter-comparison days are indicated by the dashed lines.





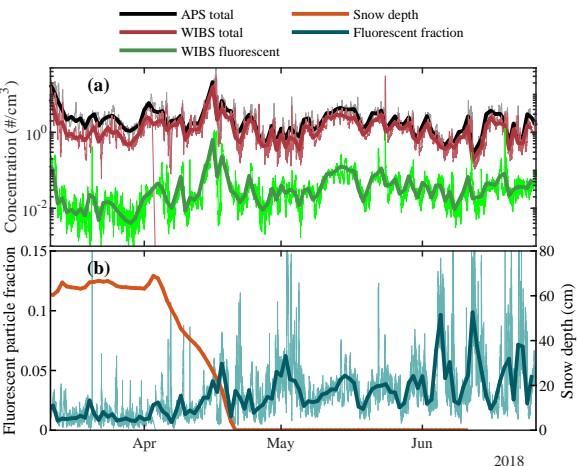

**Figure 8.** (a) WIBS fluorescent particle concentration, WIBS total particle concentration and APS total particle concentration as a function of time. (b) Fluorescent fraction (*f = WIBS fluorescent / WIBS total concentration*) and snow depth as a function of time. For both panels, the thin lines represent 10 minute averaging while the thick lines are a smoothed daily average. See §2.3.1 for more information concerning the WIBS measurement principle.

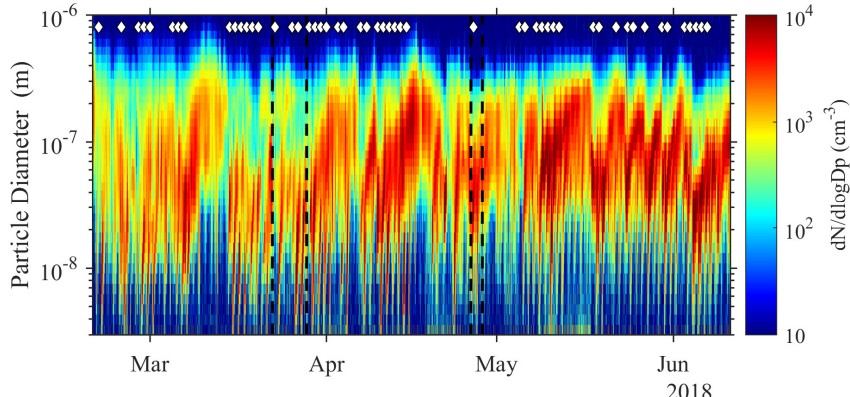

**Figure 9.** Particle number size distribution obtained from the SMEAR II DMPS for the duration of the HyICE-2018 campaign. NPF events are indicated by diamonds as in Fig. 7. Instrument inter-comparison days are indicated by the dashed lines.



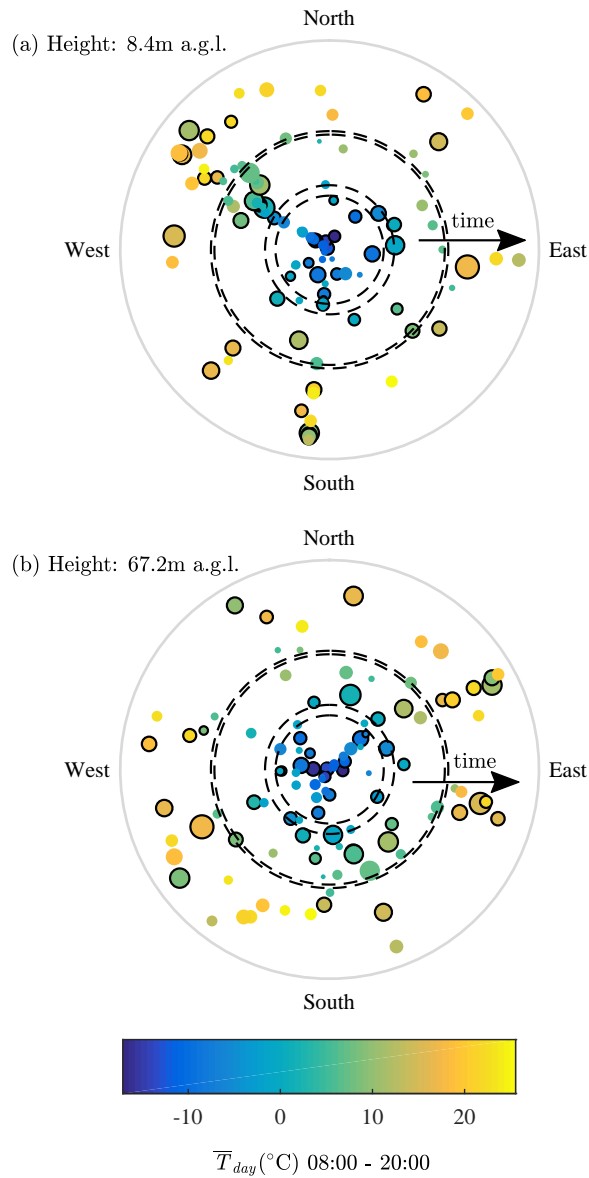

**Figure 10.** Wind roses illustrating multiple aerosol features with respect to wind direction and time. Point size and color represent the relative average daytime particle concentration (calculated from the SMEAR II DMPS and APS total concentrations) and air temperature (color bar) respectively, with days that included NPF events emphasized with black borders. Point position represents the mode of wind direction during day time, recorded at (a) 8.4 m and (b) 67.2 m (right side panel) above ground level on the mast (below and above canopy), with the radius depicting the campaign day. The inset dashed circles represent the 4 days of inter-comparison activities for INP instruments.



### 3.2 INP measurement inter-comparisons

To maximize the number of instruments that were available for the inter-comparison study, March 22, March 28, April 26 and April 28, 2018 were chosen for instrument inter-comparison. During these days, the online chambers PINC, PINCii, SPIN and PINE were operated with thermodynamic conditions close to one another and an effort was made to maximize the temporal overlap of measurements. Filter sampling for droplet freezing assay inter-comparison measurements were carried out on March 28, 2018.

### 3.2.1 Online instrument inter-comparison

A summary of the thermodynamic conditions used with the online chambers during the inter-comparison days is shown in Table 2. Note that, as mentioned earlier, both PINC and SPIN were using concentrators to increase their signal-to-noise ratio, which might lead to potential bias in their results. For PINC, the ambient INP concentration was back-calculated from the concentrated measurements, while the SPIN data presented here is the concentrated INP concentration per volume of sampled air.

**Table 2.** Thermodynamic conditions and inlet settings used for the online instrument inter-comparison studies conducted on March 22, March 28, April 26 and April 28, 2018.

| Inter-comparison date | Instrument | Lamina temperature $T_l$ or temperature $T$ | Relative Humidity with respect to water $RH_w$ | Inlet setting | Concentrator |
|---|---|---|---|---|---|
| **March 22, 2018** | SPIN | -29°C | 105% | PM$_{2.5}$ + dryer | Yes |
| | PINC | -29°C | 105% | *Morning*: Heated *Afternoon*: PM$_{2.5}$ + heated + dryer | *Morning*: No *Afternoon*: Yes |
| | PINE | -29°C | - | Heated | No |
| **March 28, 2018** | SPIN | -31°C | 105% | PM$_{2.5}$ + dryer | Yes |
| | PINC | -31°C | 105% | PM$_{2.5}$ + heated + dryer | Yes |
| | PINE | -30°C | - | Heated | No |
| **April 26, 2018** | PINCii | -32°C | 105% | PM$_{2.5}$ + heated | No |
| | PINE | -29°C | - | Heated | No |
| **April 28, 2018** | PINCii | -32°C | 105% | PM$_{2.5}$ + heated | No |
| | PINE | -29°C | - | Heated | No |

Figure 11 presents the time series of each chambers' measurements during these inter-comparison days. The PINE data is presented as a 5 point moving average to smoothen the high frequency variability, with error bars that represent 20% uncertainty in absolute INP concentrations (cf. Möhler et al., 2021). The SPIN data is generated from the difference between 15 min sampling averages and 5 min interpolated background concentrations, with error bars that represent ±1 standard deviation of the processed signal. The PINC and PINCii data is processed in an analogous manner to the SPIN data, but with sampling windows of 20 min and 15 min, respectively, and background windows of 5 and 15 minutes, respectively.

As seen in Fig. 11, there is good agreement in the INP concentrations from the PINE and PINC chambers. The SPIN chamber tends to measure lower concentrations than PINC and PINE despite the use of a concentrator, and this systematic offset is likely





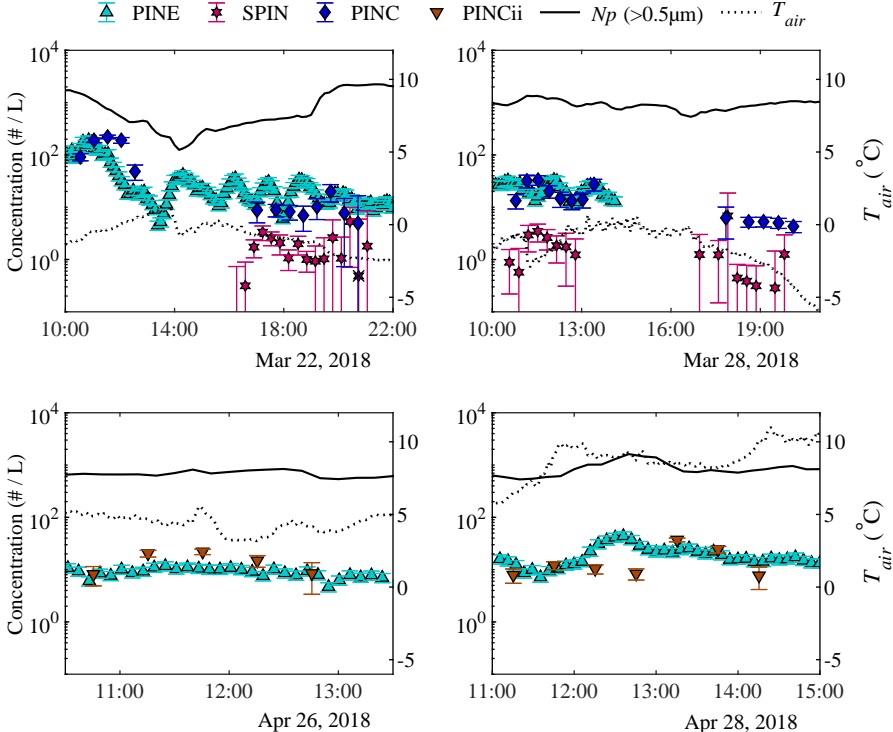

**Figure 11.** INP concentration at -29 to -32 °C as a function of time for the online INP chambers SPIN, PINC and PINE on March 22 and 28, and with PINCii on April 26 and 28, 2018. See Table 2 for more information concerning the thermodynamic conditions at which the chambers were running. Traces of aerosol number concentration, $Np$(>0.5 μm), and ambient air temperature, $T_{air}$, are plotted as solid and dotted lines, respectively. Note that the single SPIN data point that includes a black cross on March 22, 2018 is deemed to be below the level of detection, defined as the average background signal plus one standard error of the mean.

partially due to instrumental differences. In the cases presented here, the PINC data is corrected by a uniform scaling factor of 1.14 determined from well-characterized particle losses and lamina spreading measured in the instrument (Paramonov et al., 2020). Currently no simple scaling factor is available for the SPIN during HyICE-2018, and thus, no correction factor is used for the SPIN results. However, the biases in the ice-activated fraction from the SPIN chamber were discussed in Korhonen

5    et al. (2020), when the chamber was used in a separate laboratory experiment. The study shows that at approximately -31 °C, the activated fraction is biased low by a factor of ≈ 3 due to lamina spreading and particle losses, which may partly explain why SPIN measures systematically lower INP concentrations than PINE and PINC during HyICE-2018.

Figure 11 also shows very good agreement between the chambers PINE and PINCii, with INP concentrations measured within the same order of magnitude and similar trends throughout the day. The few observed deviations remain within one

10    order of magnitude and might be due to the 3 °C temperature difference between PINCii and PINE.





The aerosol number concentration for particles with an aerodynamic diameter between 0.5 and 20 μm (noted $Np$ (>0.5 μm) obtained from the SMEAR II APS is plotted for comparison (Fig. 11, solid black lines) and suggests a reasonably constant activated fraction throughout the inter-comparison days. On March 22, the aerosol number concentration gradually decreases between 10:00 and 14:00 (UTC+2), and it is interesting to observe that such variability is reflected in both the PINE and PINC data. The ambient air temperature is also represented (Fig. 11, dotted lines) and shows a normal diurnal variability.

In Fig. 12, the INP concentrations from the various chambers are superimposed onto parameterizations that predict INP concentrations. The parameterization by DeMott et al. (2010) was developed by combining observations from nine different field studies, while the parameterization by Tobo et al. (2013) proposes a modified version focusing on fluorescent biological aerosol particles. Both parameterizations use the number concentration of aerosol particles with diameters larger than 0.5 μm and the cloud temperature. In the figures presented here, the DeMott et al. (2010) and Tobo et al. (2013) parameterizations were calculated using the aerosol number concentration obtained from the SMEART II APS ($Np$ (>0.5 μm)) and a cloud temperature window between −29 and −32 °C. The additional Schneider et al. (2021) parameterization presented in Fig. 12 has been developed from long term observations conducted in Hyytiälä during and after the HyICE-2018 campaign, and relies on both ambient and cloud temperatures.

From the figures, the Tobo et al. (2013) parameterization shows the best agreement with the INP concentrations measured during the inter-comparison study. The DeMott et al. (2010) parameterization tends to underestimate the INP concentration measured by all the online chambers except for SPIN, whose uncorrected INP concentration is systematically lower than the concentration measured by the other chambers, as discussed previously. Furthermore, the Schneider et al. (2021) parameterization tends to overestimate the INP concentration measured and does not reflect the variability observed in the INP concentrations. The Schneider et al. (2021) parameterization relies on ambient air temperature, which was relatively constant over the inter-comparison days, and was developed to capture the seasonal variation of INP concentration in a time resolution of one to several days. Thus the longer time resolution inherent in the Schneider et al. (2021) parameterization might explain why it fails to reflect the daily variability observed in INP concentrations. Moreover, the parameterization was established using INP data down to -23 °C and was not tested for applications at lower temperatures; thus, deviations between the online measured and the calculated INP concentrations were to be expected. On the other hand, the majority of the data used for the DeMott et al. (2010) parameterization was collected via aircraft measurements conducted in various environments, which might explain why it tends to underestimate the INP concentrations measured at ground level. As for the good performance of the Tobo et al. (2013) parameterization, it might be related to the fact that the parameterization focuses on biological aerosol particles, which might represent an important proportion of the aerosol population in Hyytiälä (Fig. 8).

Despite the reasonable agreement with the Tobo et al. (2013) parameterization, on March 22 none of the parameterizations successfully represents the measured concentrations. Before 13:00 (UTC+2) on March 22, the INP concentrations are higher than the concentrations predicted by Tobo et al. (2013) and are more comparable to the Schneider et al. (2021) parameterization. In contrast, INP concentrations measured after 20:00 (UTC+2) on March 22 and after 17:00 (UTC+2) on March 28 decrease to below the Tobo et al. (2013) parameterization and are better represented by DeMott et al. (2010) parameterization. Therefore, although the Tobo et al. (2013) parameterization performs better than the other two parameterizations, the INP



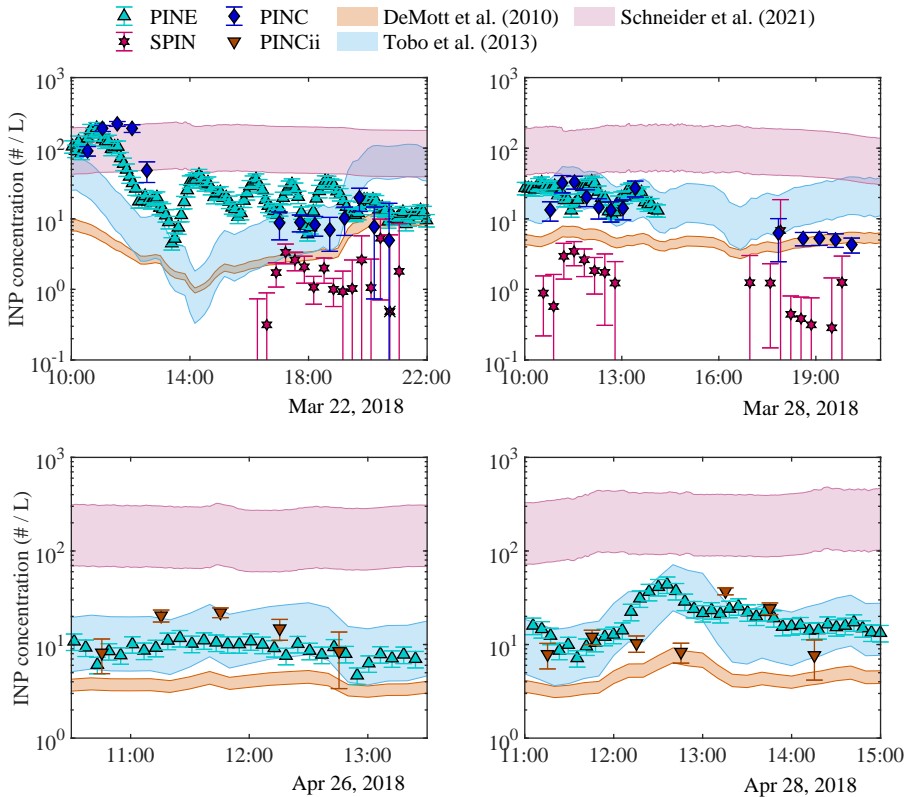

**Figure 12.** INP concentration as presented in Fig. 11 overlaid with three INP parameterizations. Both the DeMott et al. (2010) and Tobo et al. (2013) parameterizations rely on $Np$ (>0.5 μm), while the Schneider et al. (2021) parameterization relies on ambient air temperature. All parameterization envelopes represent a $-29 \geq T \geq -32\,^{\circ}$C cloud temperature window.

concentrations measured during the four inter-comparison days seem to be influenced by other factors than those included in the parameterizations. Such an observation stresses the need for new and improved understanding to better represent the INP concentrations in boreal forest environments.

### 3.2.2 Offline instrument inter-comparison

Filter sampling for the droplet freezing assay inter-comparison was conducted on March 28 and extended through the morning of March 29. Figure 13 depicts the time periods during which sampling took place. Although the filters were not collected at the exact same time, efforts were made to maximize the temporal overlap of the samples. The INP temperature spectra of the collected filters are presented in Fig. 14. Although INSEKT detects INPs at temperatures up to 5 °C warmer than μL-NIPI, the INP temperature spectra show good overlap between the two techniques and strong temporal agreement. An increase in the INP concentration is observed for the filters that were collected exclusively during daytime hours, with no clear bias between





the morning and the afternoon. As reflected in Fig. 11, the aerosol number concentration remained nearly constant on that day, while other meteorological parameters, like the ambient air temperature, were fairly stable with expected diurnal variability.

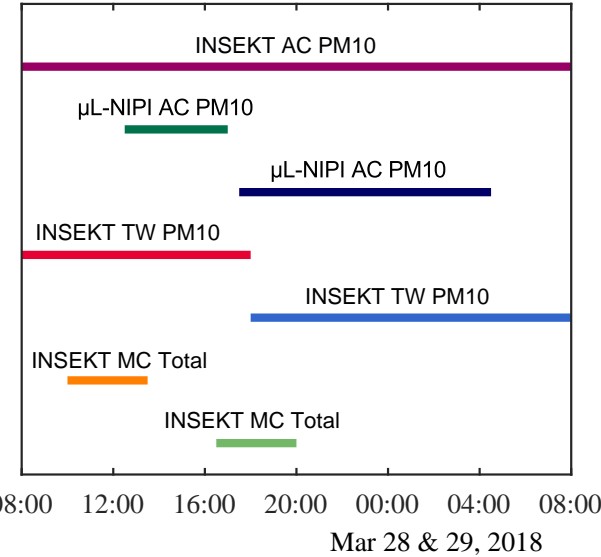

**Figure 13.** Timeline of the filter sampling carried out on March 28, 2018. The droplet freezing assays used for the INP analysis (INSEKT; µL-NIPI), the sampling locations (MC = main cottage; AC = aerosol cottage; TW = tower) and the inlets used during sampling (Total, PM10) are indicated. The colored bars indicate the various sampling time windows and correspond to the INP temperature spectra presented in Figs. 14 and 16.

It should be noted that dilutions were used to extend the INSEKT detection range to lower temperatures. For the measurements presented in Fig. 14, the data points between -12 and -18 °C are coming from the non-diluted aerosol solution while the data points at lower temperatures are coming from a solution that was diluted by a factor of 10. Although the decreasing "step" we observe between the series of data points is nonphysical, we nevertheless decided to present all data points here as measured. Moreover, when considering the spectra in the range of the error bars, the data points overlap and the INP concentration still increases with the decreasing temperature.

In Fig. 15, the INSEKT and µL-NIPI methods are directly compared for samples selected with as much temporal overlap as possible. The results are very similar, and a good agreement is observed over the entire range of temperatures. The primary source of deviation in the agreement between the two methods is shown in Fig. 15c, which is expected due to a shorter temporal overlap in the sample collection for these two filters. More specifically, the data shown in Fig. 15c was obtained from one filter collected between 12:30-17:00 (UTC+2) for by µL-NIPI and one filter collected from 16:30-20:00 (UTC+2) for INSEKT, representing only 30 minutes of overlap between the measurements.



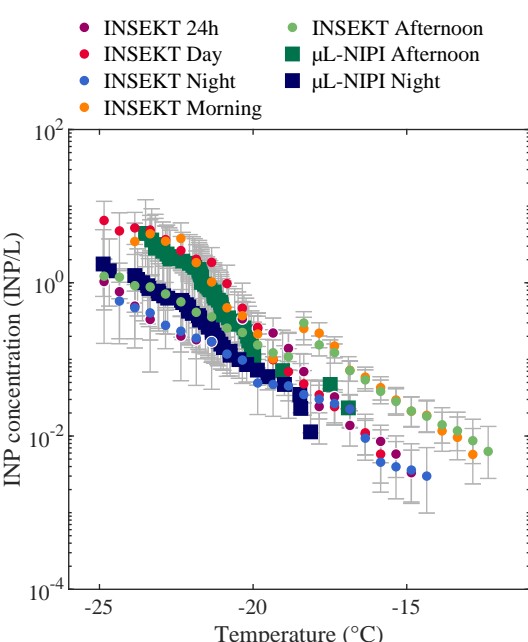

**Figure 14.** INP temperature spectra measured for each filter collected on March 28 as indicated in Fig. 13. For the INSEKT data, the error bars represent the statistic as well as the systematic error of the INSEKT assay. More details related to the calculation of these error bars is given in Schneider et al. (2021). For the µL-NIPI data, the error bars were calculated using the Poisson Monte Carlo procedure as described in Harrison et al. (2016)

.



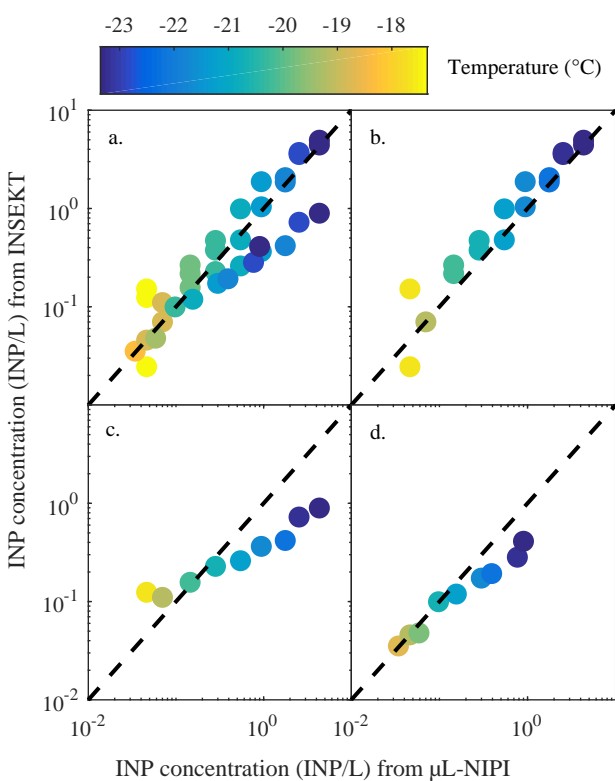

**Figure 15.** INP concentration measurements from INSEKT and μL-NIPI compared to one another for March 28. Point color represents the ice nucleation temperature. The panels include, (a) the full measurement set (b) INSEKT day and morning (red and orange bars in Fig. 13) versus μL-NIPI afternoon measurements (dark green in Fig. 13) (c) INSEKT afternoon (light green in Fig. 13) versus μL-NIPI afternoon measurements (dark green in Fig. 13) and (d) INSEKT night (dark blue in Fig. 13) versus μL-NIPI night measurements (light blue in Fig. 13). In each case the dashed line represents the 1:1 line.





### 3.2.3 Inter-comparison summary

In Fig. 16, the measurements from all INP instruments sampling on March 28 are presented in the form of an INP temperature spectra. The data points from the online INP measurement systems SPIN, PINC and PINE were obtained by averaging the measured INP concentrations over the entire day, and the error bars now represent the standard deviation of the processed data

shown in Fig. 12. The selected INP parameterizations are also depicted in the figures, where the shaded regions represent one standard deviation above and below the daily mean parameter values of each parameterization for March 28. As expected, in this context the Schneider et al. (2021) parameterization performs much better at warmer temperatures (between -12 and -25 °C, the temperatures for which the parameterization was established). However, it only agrees with part of the INP temperature spectra obtained from the INSEKT and µL-NIPI methods. The Tobo et al. (2013) parameterization shows the best agreement

with the INP concentrations from both the online and the offline INP measurement systems. Conversely, the DeMott et al. (2010) parameterization performs poorly at representing the online systems with the exception of SPIN, and it does not capture the ice nucleation behaviour observed by the droplet freezing assays at warmer temperatures. However, the DeMott et al. (2010) parameterization is based on various measurements conducted in very different environments where the majority of the data were collected via aircraft measurement, and thus bias can be expected. The performance of the various parameterizations

highlights the need to critically assess the validity range(s) of such empirical parameterizations, especially when they are suggested for utilization in predictive models.

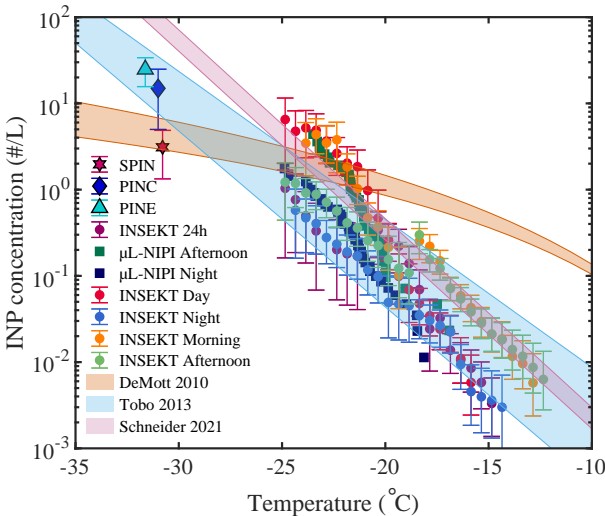

**Figure 16.** INP temperature spectra from all the instruments running March 28. Error bars assigned to online measurement techniques represent the standard deviation of the processed data shown in Fig. 12. The DeMott et al. (2010), Tobo et al. (2013) and Schneider et al. (2021) parameterizations are replotted with the envelopes now indicative of the average $Np$ (>0.5 µm) $\pm 1$ standard deviation for DeMott and Tobo, and the average air temperature $\overline{T}_{air} \pm 1$ standard deviation for the Schneider parameterization.





## 4    Conclusions

In this measurement report we introduced the HyICE-2018 campaign conducted at the SMEAR II research station in the boreal
forest of Hyytiälä, Finland. The campaign took place from February to June 2018 and utilized the infrastructure of the SMEAR
II station with additional instrumentation focusing on INP measurements. The main objectives of the campaign were to

• quantify and characterize INPs in a boreal environment within different thermodynamic conditions

       • examine the seasonal variability of the INP concentrations and

       • study the vertical distribution of INPs above the boreal forest.

Several days during the campaign were selected to inter-compare INP measurement systems in a field setting. The instru-
mentation included three CFDCs (PINC, PINCii and SPIN) and one expansion chamber (PINE) for online measurements of
INPs, and two droplet freezing assays (INSEKT and μL-NIPI) for offline analysis of INPs collected on filters. Results from the
online measurements show that, despite the differences in setups and measurements settings, there is a good agreement between
the PINE and PINC chambers, and between the PINE and PINCii chambers. The chambers detect INP concentration in the
same order of magnitude and show similar trends throughout the inter-comparison days. The SPIN chamber tends to measure
lower concentrations than the other chambers, likely in parts due to instrumental biases. Results from the offline measurements
show reasonable agreement between the two droplet freezing assays INSEKT and μL-NIPI. Although INSEKT tends to detect
INPs at temperatures up to 5 °C warmer than μL-NIPI, the INP temperature spectra show good overlap between the two tech-
niques and good temporal agreement. The direct comparison of the INP concentrations obtained from both techniques shows a
good agreement, and the results illustrate the importance of longer sampling durations and longer temporal overlap to improve
agreement.

The measurements from the inter-comparison days were compared to three existing parameterizations. The DeMott et al.
(2010) parameterization tends to underestimate the observed INP concentrations while the Schneider et al. (2021) parameteri-
zation tends to overestimate the INP concentration and does not capture the daily variability observed in the INP concentration.
The parameterization by Tobo et al. (2013) shows the best agreement with the measured INP concentrations, which is attributed
to the fact that it focuses on biological aerosol particles which might represent an important proportion of the aerosol population
in the boreal forest of Hyytiälä. The parameterization was developed from measurements conducted in a mid-latitude ponderosa
pine forest ecosystem in Colorado, US, and it is interesting to see that its results are consistent with some of the measurement
obtained during the HyICE-2018 campaign. Although more analyses would be required to compare the parameterization to the
INP concentrations measured during HyICE-2018 in a more systematic way, the results suggest that the sources of INPs in the
boreal forest might be comparable to those in a northern-American forest. On the other hand, despite the good agreement with
the Tobo et al. (2013) parameterization, there is one day of inter-comparison when none of the parameterizations successfully
represent the measured concentrations. This suggests that, on that day, the INP concentration might be influenced by factors
that are not included in the parameterizations used here, and that continuous efforts are required to improve parameterizations
to better represent the INP concentrations from boreal forest environments.



Although not all the objectives written above were addressed in this paper, a number of results from the campaign will be or have already been addressed in separate contributions for the Copernicus Special Issue "Ice nucleation in the boreal atmosphere". So far, the special issue includes an examination of the PINC data during winter 2018, which showed that no persistent local sources of INPs could be identified and which postulated that the INPs detected were the result of long-range

5  transport and dilution of INPs sourced far from the measurement site (Paramonov et al., 2020). Another study from Schneider et al. (2021) presented a year-long record of INP concentrations measured with INSEKT and showed a clear seasonal cycle of INP concentrations and INP types in the boreal forest, most likely driven by the abundance of biogenic aerosol. Forthcoming studies will explore atmospheric vertical profiles of INPs, INP sources and transport modeling, plausible links between INP concentrations and new particle formation (NPF) events, ice nucleation activity of collected biological materials such as plants

10  and fungi, and also utilize the full suite of available measurements to illuminate non-obvious processes that feedback on INP concentrations and properties.



**Appendix A**



**Table A1.** Overview of the SMEAR II instrumentation running throughout the HyICE-2018 campaign. [Part 1]

| Quantity/ property measured | Method/Instrument | Time resolution | Location |
|---|---|---|---|
| **Atmospheric aerosol measurements** | | | |
| Aerosol particle and ion size distribution | Particle Size Magnifier (PSM; Airmodus model A10) (1- 3 nm) | 1 s | Tower (35 m) and container (2 m) |
| | Twin Differential Mobility Particle Sizer (DMPS) (3 - 1000 nm) | 10 min | Tower (35 m) and hitu-hut (8 m) |
| | Aerodynamic particle sizer (APS; TSI model 3320) (0.5- 20 μm) | 10 min | Hitu-hut (5 m) |
| | Neutral Cluster and Air Ion Spectrometer (NAIS; Airel)(0.8-42 nm for ions) | 1 min | Tower (35 m) and hitu-hut (2 m) |
| Air ion mobility distribution | Balanced Scanning Mobility Analyzer (BSMA) (0.4-7.5 nm) | 10 min | Rea-hut (2 m) |
| Clusters composition and concentration | Chemical Ionization Atmospheric Pressure Interface Time-of-flight mass spectrometer (CI-API-TOF; Tofwerk AG) | 1 s | Container (2 m) |
| Cloud Condensation Nuclei (CCN) | CCN counter (DMT model CCN-100) | 1 sec | Hitu-hut (8 m) |
| $PM_{10}$ Particle mass concentration | Particulate Detection Monitor (Thermo Scientific; model 5030 SHAR Monitor) | 1 min | Hitu-hut (5 m) |
| Black Carbon (BC) particle mass concentration | Multi Angle Absorption Photometer (MAAP; Thermo Scientific) | 1 min | Hitu-hut (5 m) |
| | Aethalometer (Magee Scientific model AE-33-7) | 10 min | Hitu-hut (5 m) |
| Aerosol optical properties | Continuous Light Absorption Photometer (CLAP) | | Hitu-hut (5 m) |
| | Nephelometer (TSI model 3563) | 10 min | Hitu-hut (5 m) |
| | Sun photometer (Cimel model CE-318) | | |
| Aerosol chemical composition | Aerosol Chemical Speciation Monitor (ACSM; Aerodyne Research Inc. USA) | 30 min | |
| | Organic Carbon/Elemental Carbon Analyser (OCEC ; Sunset Laboratory) | 4 hours | |
| | Time-of-Flight Aerosol Mass Spectrometer (L-ToF-AMS) | | |
| Size classified mass collection of aerosol particles | $PM_{10}$ impactor (DEKATI) | 3 days | Hitu-hut (5 m) |
| $PM_{10}$ Heavy metals | Filter sampler (MCZ model MicroPNS S7) | 7 days | |
| $PM_{10}$ Main ions | Filter sampler (3-stage EMEP) | 7 days | |



**Table A2.** Overview of the SMEAR II instrumentation running throughout the HyICE-2018 campaign. [Part 2]

| Quantity/ property measured | Method/Instrument | Time resolution | Location |
|---|---|---|---|
| **Meteorology** | | | |
| Ambient air temperature | Ventilated and shielded sensors (Pt-100) | 10 min | Rea-hut (2 m) |
| | Rotronic RS12T / RS24T, HygroMet MP102H-530300, HygroClip2 (HC2-S3) | | Mast (125 m) |
| Ambient air pressure | Druck DPI 260 barometer | 1 min | Mast (0 m) |
| | Vaisala PTB210 barometer | 10 sec | Mast (30 m) |
| | Metallic aneroid capsule capacitance, Vaisala PA-11 | 1 min | Field (2 m) |
| Relative humidity and dew point temperature | Vaisala dew point transmitter | 10 s | Field (1.5 m) |
| | Hygroclip2 probe | | Tower (35 m) |
| | Rotronic humidity and temperature sensor MP102H | 5 sec | Tower (35 m) |
| | chilled mirror dew point monitor | 1 min | 18 m tower (16 m) and mast (23 m) |
| | EdgeTech Model 200M Meteorological System | 1 min | Tower (35 m) |
| Wind speed and direction | Thies Ultrasonic Anemometer 2D | 1 min | Mast (8.4, 16.8, 33.6 and 67.2 m) |
| | Metek uSonic -3 Anemometer | 0.1 s | Mast (67.2 and 125 m) |
| | Gill 3D sonic anemometer | 0.1 s | Field (1.5 m) |
| | Vaisala 2D sonic anemometer | 1 s | Field (3 m) |
| Precipitation | Vector ARG-100 tipping bucket rain gauge | 1 sec | Tower (35 m) and catchment (2m) |
| | Bucket weighing, OTT Pluvio2, 200 and 400 sq.cm | 1 min | Field (1.5 and 4 m) |
| Surface precipitation, particle size distribution | Optical disdrometer, OTT Parsivel2 | 1 min | Field (4 m) |
| | 2D video disdrometer (particle video imaging) | 1 min | Field (4 m) |
| | Particle video imaging, PIP, NASA | 1 min | Field (1 m) |
| | Particle holographic imaging, SAKU III | 1 min | Field (2 m) |
| Snow depth | Manual measurement at seven locations | 7 days | Catchment |
| | Jenoptik SHM30 optical snow depth sensor | 1 min | Field |



**Table A3.** Overview of the SMEAR II instrumentation running throughout the HyICE-2018 campaign. [Part 3]

| Quantity/ property measured | Method/Instrument | Time resolution | Location |
|---|---|---|---|
| **Remote sensing** | | | |
| Vertical profile of hydrometeors and wind profile Cloud and boundary layer height | 94 GHz FMCW Doppler Cloud Radar (RPG-FMCW-94-DP) | 3 s | Field |
| | Microwave radiometer - Humidity And Temperature PROfiler (RPG-HATPRO) | 3 s | Field |
| | HALO Photonics Doppler lidar | 16 s | Roof |
| | Ceilometer | 16 s | Hitu-hut |
| **Solar and terrestrial radiation** | | | |
| **Gas measurements** (includes $CO_2$, $H_2O$, $CH_4$, CO, $O_3$, $SO_2$, $NO_X$, VOC measurements) | | | |
| **Fluxes measurements** (includes $CO_2$, $H_2O$, $CH_4$, aerosol particles, momentum and heat flux measurements) | | | |
| **Soil and water balance** (includes snow and water collection; soil temperature, water content, matric potential and heat flux measurements; measurements of $CO_2$, $CH_4$ and VOC concentrations and fluxes in the forest floor) | | | |
| **Forest ecophysiology and productivity** (includes exchange of gas by shoots and stems measurements; sap flow measurement; litter collection; etc.) | | | |





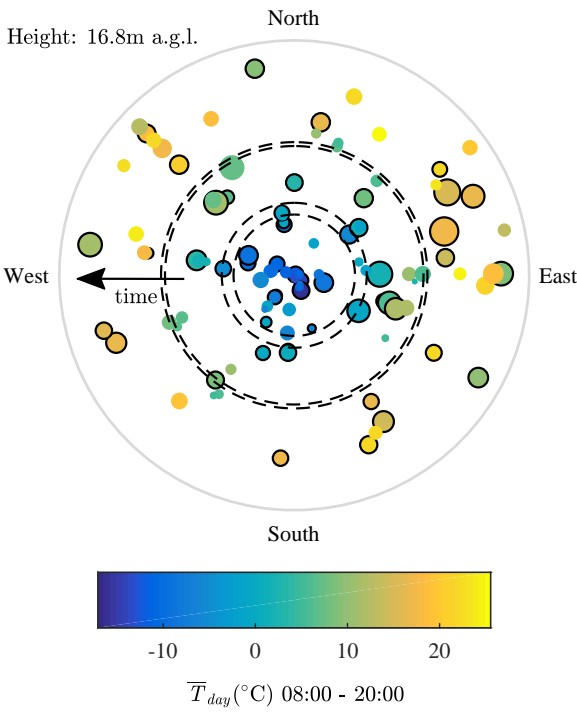

**Figure A1.** Wind rose plot illustrating multiple aerosol features with respect to wind direction and time. Point size and color represent the relative average daytime particle concentration (calculated from DMPS and APS total concentrations) and temperature (color bar) respectively, with days that included NPF events emphasized with black borders. Point position represents the mode of wind direction during day time (recorded at 16.8 m above ground level on the mast) with the radius depicting the campaign day. The inset dashed circles represent the 4 days of inter-comparison activities for INP instruments.



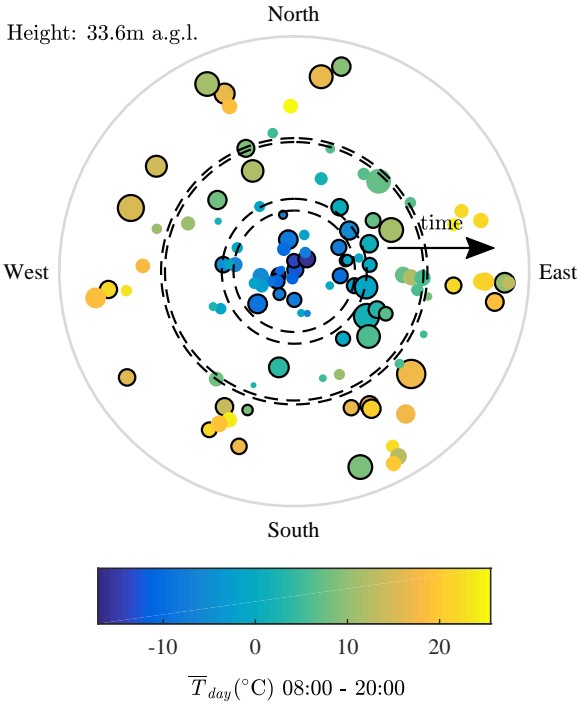

**Figure A2.** Wind rose plot illustrating multiple aerosol features with respect to wind direction and time. Point size and color represent the relative average daytime particle concentration (calculated from DMPS and APS total concentrations) and temperature (color bar) respectively, with days that included NPF events emphasized with black borders. Point position represents the mode of wind direction during day time (recorded at 33.6 m above ground level on the mast) with the radius depicting the campaign day. The inset dashed circles represent the 4 days of inter-comparison activities for INP instruments.





*Data availability.* The aerosol, traces gas and meteorological data are available at the SmartSMEAR data repository (https://avaa.tdata.fi/web/smart, Junninen et al. 2009). Contact of the original data contributors can be requested from atm-data@helsinki.fi. The INP data presented in this study are available at https://doi.org/10.5281/zenodo.5141574. Other data are available upon request from the corresponding authors. The data discussed and presented from PINC are also included in (Paramonov et al., 2020) and are available at https://doi.org/10.3929/ethz-b-

5   000397022.

*Author contributions.* OM, MK, TP, JD and DM initiated and planned the HyICE-2018 campaign. ZB and JD largely coordinated and oversaw the campaign on-the-ground with the help of the permanent SMEAR II staff. ZB, DC, MPA, SDvD, PH, KK, JL, MP, JS, FV, YW, NSA, BB, DB, MID, ADH, PH, KH, LL, ML, AM, MP, GCEP, PP, UP, TS and NSU conducted measurements during the HyICE-2018 campaign. ZB, DC, MPA, PH, KK, MP, JS, FV, and YW performed the data analysis of their respective instruments and took part in

interpretation of the results. JPDA and EG provided the PFPC and EG setup the instrument at the station. EST, JPDA, DHB, ZAK, JK, AV, MK, BJM, TP, OM and JD participated as supervisors of the HyICE-2018 campaign. ZB and EST wrote the manuscript and with the help of DC constructed the figures. MB helped writing and revising the manuscript. All authors reviewed the manuscript.

*Competing interests.* The authors declare no competing interests.

*Acknowledgements.* This project received funding from the European Union's Horizon 370 2020 research and innovation programme under

grand agreement No 654109 and 739530 and Transnational access via ACTRIS-2 HyICE-2018 TNA project. The work of University of Helsinki was supported by the Academy of Finland Centre of Excellence in Atmospheric Science (grant no. 307331) and NANOBIOMASS (307537), ACTRIS-Finland (328616), ACTRISCF (329274) and Arctic Community Resilience to Boreal Environmental change: Assessing risks from fire and disease (ACRoBEAR, 334792) Belmont Forum project. In addition, the work of University of Helsinki was financially supported by European Commission through ACTRIS2 (654109) and ACTRIS-IMP (871115) and ACTRIS2 TransNational Access and

through integrative and Comprehensive Understanding on Polar Environments (iCUPE, 689443), ERA-NET-Cofund and by University of Helsinki (ACTRIS-HY). The work of the KIT Institute for Meteorology and Climate Research (IMK-AAF) was supported through the Research Program "Atmosphere and Climate (ATMO)" of the Helmholtz Association and by the KIT Technology Transfer Project PINE (N059). EST and DC have been supported by the Swedish Research Councils, VR (2013-05153, 2020-03497) and FORMAS (2017-00564) and the Swedish Strategic Research Area MERGE. BJM and MPA acknowledge the European Research Council, ERC, MarineIce 648661

for funding. PH and JK acknowledge the funding from the Arctic Academy Programme , "ARKTIKO" of the Academy of Finland under grant No 286558, and PH from the Maj and Tor Nessling Foundation. NSU acknowledges the support of the Alexander von Humboldt Foundation, Germany (1188375). MP, SDvD and ZAK acknowledge the funding from the European Union's Horizon 2020 research and innovation programme (under the Marie Sklodowska-Curie grant agreement no. 751470, "ATM-METFIN" and grant agreement no 654109) and the European Union Seventh Framework Programme (FP7/2007-2013; grant agreement no. 262254). EG and JPDA acknowledge the

support from the NSERC (grant no. RGPIN-2017-05972) and EG from the University of Toronto Centre for Global Change Science Graduate





Student Research Award. NSA and DHB acknowledge the funding from the Academy of Finland under the Postdoctoral Researcher Grant 309570 to N.S.A.

Furthermore, the authors would like to gratefully acknowledge Janne Levula, Matti Loponen, Heikki Laakso, Turo Salminen and the rest of the technical staff of the Hyytiälä Forestry Field Station for their expertise, hard work and willingness to help throughout the HyICE-2018 campaign. Erkki Järvinen and the pilots at Airspark Oy are acknowledge for operating the research airplane. Simo Hakala is acknowledged for his help with the new particle formation events classification. Hanna Danielsson is acknowledged for her significant help with creating Figure 1.



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
