# Peer review of "Measurement report: Introduction to the HyICE-2018 campaign for measurements of ice nucleating particles and instrument inter-comparison in the Hyytiälä boreal forest"

_Atmospheric Chemistry and Physics, 2021_

## Referee Comment (RC1)

**Review of "Measurement report: Introduction to the HyICE-2018 campaign for measurements of ice nucleating particles in the Hyytiälä boreal forest" by Brasseur et al.**

In this paper, the authors depicted the scope and detailed introduction of HyICE-2018 campaign, as well as inter-comparisons for both online and offline ice nucleating particle (INP) concentration measurement instruments based on 4-day data. The paper fits the scope of ACP, and is properly structured and well-written. The paper needs to be reviewed again after the authors address the reviewer's concerns below appropriately.

**Major comments**

1. The paper uses too many vague and intuitive descriptions, such as "good" and "poorly". Even though it is a measurement report, it's still a scientific contribution to the community. Therefore, more scientific and quantitative descriptions are required.

2. There are too many figures than actually needed with repeated information. Some of them are barely mentioned in the text. Please carefully reconsider and prepare the figures.

**Minor comments**

Title: Please explicitly include "instrument inter-comparison" in the title and clarify the INP measurement condition, i.e., mixed-phase cloud formation condition, here and in abstract, and delete the full stop at the end.

P2L1: Even though not all INP instruments were run simultaneously, it would still be helpful to list the INP instrumentation engaged in this study here instead of using "additional instrumentation for measuring INPs" to give the readers more information on what type of INP instruments were used. Besides, please give the full name of SMEAR according to ACP requirement.

P2L14: It would be more logical to add a statement about ice formation pathways in the atmosphere (homogeneous and heterogeneous freezing) before introducing INP and heterogeneous ice nucleation.

P3L14-L15: Please specify the experiment condition (temperature and supersaturation with respect to water) of the organic IN references. Are they relevant to the experiment condition of this study?

P4L4: Please define a.s.l.

P4L8: Please rephrase the sentence "Multiple towers include…" to "There are several towers…, including …".

P5L11-L12: After heating the APS inlet, what did the authors do with the water vapor in the sampling flow? Will the water vapor condense inside the sampling line?

P5L14: Please change "operational" to "operating". Also change it on P9L15.

P5L16: Please use consistent time format throughout the paper.

P5L17-L20: This sentence is irrelevant, please remove it.

P6L1: What makes the sampling "augmented"?

P6L12: Please rephrase to "offline IN activity/concentration measurements of particles collected on filters". The original sentence reads as if the authors characterized INPs using the two DFA instruments, which is somehow beyond the instruments' capability.

P6L15: Please define PINC, PINCii, and SPIN.

P7L5: Please add a comma before "some particles will…".

P7L9: I suggest to replace "frozen and unfrozen particles" to "ice crystal and water droplets".

P8L5-L6: Please rephrase "sampling intervals of 5 to minutes…". Did the three CFDCs use different

background sampling time?

P8L11-L12: How were the concentrations during background and measurement sampling window determined? Were they converted from OPC counts and averaged over time? If so, pleas indicate the deduction and average before "concentrations".

P8L24-L25: What's the influence of such a large inlet flow rate compared to the small sampling flow rate? Is it possible that the vacuum pump driving inlet flow sucks from the instruments?

P8L26: What's the $RH_w$ range of the sampling stream at the dryer outlet?

P9L2: What is the second enrichment factor and how was it determined? Was it for the whole particle size distribution spectrum or for each size?

P9L16: Why is there a 1 °C lamina difference between PINC and PINCii to produce comparable results? How many thermocouples were used to determine PINC and PINCii wall temperature, respectively?

P9L17: What is the $RH_w$ when the sampling air enter PINCii? Is it possible that the water vapor in the sampling stream condense and affect the measurement and background results at PINC inlet?

P10L6: If the SPIN design in this study is the same as Garimella et al. 2016, then the statement on P7L4 "drawn through the center of the chamber" should be revised, since SPIN lamina does not lie along the chamber centerline.

P10L11: Did the authors characterize the VI concentrator? What's the size-resolved enrichment factor? Please add references here. Please also add "(more details about the VI concentrator are provided below)" here.

P11L2: What is the exact concentration for particles larger than 1.3 μm based on APS data?

P11L3: Why are these large particles more pronounced? Please add references associated with the aerosol types presented in this study.

P11L3-L5: Did the authors measure particle size distribution before and after the VI concentrator with APS?

P11L10: $RH_w$ = 110% seems to be above the water droplet survival line at -36 °C and -32 °C.

P11L11-L13: The particle concentration varies greatly from time to time in this study according to Fig. 7. It is common practice to fix CFDC lamina condition during field studies because it usually takes several minutes to change CFDC lamina $S_i$ and temperature and re-establish equilibrium. Would the frequent alternation of both temperature and saturation in this study introduce more uncertainty? Are these results obtained with changing ambient condition and lamina condition at the same time comparable with each other? These descriptions might be irrelevant to the focus of this paper, i.e. inter-comparison of different instrumentation. So the author might just remove them.

P11L19: Please format the reference.

P11L23-L25: Will this air pumping process through OPC change aerosol population, and therefore INP concentration, in the chamber? Do the authors imply that the aerosols inside PINE chamber distribute homogeneously under changing pressure?

P12L5: Please clarify the particle transmission efficiency here and in the caption of Fig. 6 are for the sampling system upstream PINE rather than the PINE chamber itself to avoid misunderstanding.

P12L15-P13L9: Reads like introduction and adds little to the topic. Also between P13L9 and L10, do the authors mean one complete temperature spectrum in one test run?

P13L19-L20: Will the cleaning process change filter IN activity?

P13L30: How were the collected aerosols washed off and suspended? Mechanically or by supersonic? Please

elaborate.

P13L31: Please define "PCR".

P13L35: Please add references for the camera and software if it were developed by KIT. Otherwise, please clarify the production company.

P14L6-L7: It would be better to use "100 °C water bath" here. Please also elaborate what container was used to hold the suspensions in the boiling water.

P14L22: Please elaborate.

P14L30: Please remove "and INPs". These instruments can only provide vertical profile of ambient aerosols.

P15L13: What's the relationship between soot and black carbon?

P15L21-L22: What's the transition efficiency of the total aerosol inlet? Will the $PM_{10}$ inlet prevent large bio-aerosol from entering WIBS? Please elaborate more here.

P15L25-L27: What sampler did the authors use? Please add references.

P15L27-L30: Please add references for the cultivation process.

P16L12: The parenthesis should appear after "sensors".

P16L10: Please specify the snow depth and temperature range during winter instead of the vague description "deep snow cover and cold temperature".

P16L17: See the comments on these figures.

P21L14: Is there a reason why does PINE exhibit such frequent high variability? Is it valid to average INP concentration measured at different time (~30 min in this case)? Please elaborate.

P21L14: Could the authors change the 20% uncertainty to ±1 standard deviation for PINE results to be consistent with the rest of the text?

P21L19: The trends of INP concentration for PINE and PINC during 18:00-20:00, Mar. 22, and around 11:30, Mar. 28 are opposite. Please explain why does this happen.

P21L10: By a factor of what is the SPIN measured INP concentration lower than that measured by PINE and PINC?

P22L8-L10: The trends between 12:00-13:00, Apr. 28 are not "very good" and "similar" even the authors attribute the deviations to different experiment temperature choice. Besides, is there a particular reason for the different $T_1$ and T choice?

P23L5: Why is the temperature information repeated again and again? Is there any finding in relevance to the daily and long-term temperature variation?

P23L25-L26: Does that mean ground-level measurements tend to over-present realistic INP concentration in tropopause?

P24L2-L3: Given the vast quantities measured in this field campaign, the readers would expect the authors to propose a novel parameterization applicable to boreal forest environment. This would add more to the paper and to the community.

P24L7: What measures did the authors take?

P25L8: What does "…detects INPs at temperatures up to 5 °C warmer…" mean? Does it mean that INSEKT has wider measurement temperature range?

P25L9-L10: What caused the INP concentration increase with the relatively stable aerosol population and meteorology condition?

P25L3-L8: It seems like the dilution process modified the IN active component in the suspension and exhibits a discontinuous drop in Fig. 14. The readers would expect more physicochemical analysis on the aerosol type and abundance in the suspension, rather than a vague and plain statement of "in the range of error bars".

P25L13: Please remove "by".

P25L13-L14: What factors caused the large deviation between different sampling time? The authors present diurnal meteorology and particle concentration profile with no analysis and investigation.

P28L5-L6: How was one standard deviation determined for each parameterization? The content in Fig. 16 caption should be elaborated more in the main text rather than not mentioned at all.

P28L8-L14: Please quantify "good agreement" and "poorly" since this is a scientific paper. The authors may also consider tune down the conclusion here since it is drawn based on one single data point for online instruments.

P28L9-L10: It seems like that the Tobo et al. (2013) parameterization overlaps more because it has larger uncertainty based on the description and figure.

P29L15-L16: Again, what does "…tends to detect INPs at temperatures up to 5 °C warmer…" mean?

P29L16-L18: Too many vague descriptions. Please quantify "good".

P29L20-L23: Again, please tune down the conclusion here, since it is drawn based on very limited data.

P30L1-L11: Reads like introduction.

Fig. 1:

- The legend text size is too small.

Fig. 2:

- Would it be cleaner to add a column in Table 1 and include the sampling period? The daytime average temperature information appeared again in Fig. 7 and Fig. 10. Do the authors intend to show the correlation between temperature and what?

Fig. 3:

  Panel a:

- Please specify "main chamber" and "evaporation section" in the plot.

- Supersaturation > 1 means $RH_i$ > 200%. I assume the authors use $S_i$ to present saturation with respect to ice, rather than supersaturation here? If so, the definitions on P3L26 and P7L5 should also be modified.

- It would be better to draw background and measurement sampling periods (green and pink) to scale (e.g., 5 min background to 15 min measurement).

  Panel c:

- Please align letter "T" with the arrow on the top right corner.

  Panel d:

- The legend color of Expansion Chamber doesn't match (brighter red) with the plot.

Caption:

- L5-L6: Add a dash between "INP" and "temperature".

Fig. 4:

- Please annotate the circled cross mark in the scheme. For example, it could be a virtual valve.

Fig. 5:

- Could the MFCs on the right part of this figure function normally in the setup? Where does the MFCs working pressure difference come from? Is there a needle valve to choke the external pump at the Excess out (free air)?

Fig. 6:

- It adds little to the paper. Could it be moved to appendix?

Fig. 8:

- The colors and legend position for Snow depth and WIBS total are confusing. It would be better to indicate panel (a) and (b) beside the legends.

Fig. 7-9:

- Recombination of the same experiment data set. Could it be reduced to two figures? For example, Fig. 9 could be a panel of Fig. 7.

Fig. 10:

- This figure is not self-explanatory. Missing legends and too much information placed in the caption.

- It adds little to the main text. There is just one statement about this figure betweeen P17L18-P17L20. Therefore, it might better be moved to appendix.

Fig. 11:

- Caption L2: Please replace "with PINCii" with "for PINE and PINCii" to avoid misunderstanding.

- Would it be possible to incorporate inlet information listed in Table 2 in this figure? That way the readers can gain more knowledge regarding the difference between heating and inlet cut-off setups.

Fig. 12:

- Why don't the authors combine this figure with Fig. 11?

- If the authors select a temperature window of -29 ℃ to -32 ℃ here to cover the experiment temperature selected in this study, it would be more reasonable to draw two separate lines for -29 ℃ and -31℃ for each parameterization, and draw a shading representing ±1 standard deviation using different particle concentration or ambient temperature for the predicted INP concentration.

Fig. 11 and 12:

- Since this paper concentrate on instrument inter-comparison, the readers would expect a plot like Fig. 15.

Fig. 13:

- It might be more clear and straightforward to convert this figure to a table.

Fig. 14:

- The error bar color needs to be changed for either instrument.

- Why is there a drop of INP concentration around -18 °C for both instruments, even though only INSEKT used the diluted suspension?

Fig. 15:

- It might be better to use different symbols, rather than colors, to denote different measurement time.

Fig. 16:

- Add "on" before "March 28" in the caption.

Table 1:

- A list of abbreviations should be given somewhere.

- The sampling locations for INSEKT and μL-NIPI are confusing. Are the filters used for both instruments from all three locations?

---

## Author Response (AR1)

**Reviewer Response for "Measurement report: Introduction to the HyICE-2018 campaign for measurements of ice nucleating particles in the Hyytiälä boreal forest" by Brasseur et al.**

We would like to thank the reviewers for their constructive and useful comments, which have illuminated several areas for improvement and clarification within our manuscript. We try to answer each reviewer question and utilize the suggestions in order to improve the manuscript and have prepared a revised manuscript accordingly. Below we explicitly address and/or point to changes in the text that address each of the items raised in the comments. The responses are presented in the context of the Reviewer comments (reprinted in normal text), where our responses are presented in the green-shaded text. To best illustrate the extent of the changes to the text the output of a latex difference file is also attached.

**Responses to Anonymous Referee #1 (RC1)**

**Review of "Measurement report: Introduction to the HyICE-2018 campaign for measurements of ice nucleating particles in the Hyytiälä boreal forest" by Brasseur et al.**

In this paper, the authors depicted the scope and detailed introduction of HyICE-2018 campaign, as well as inter-comparisons for both online and offline ice nucleating particle (INP) concentration measurement instruments based on 4-day data. The paper fits the scope of ACP, and is properly structured and well-written. The paper needs to be reviewed again after the authors address the reviewer's concerns below appropriately.

We thank the referee for their thoughtful comments and feedback. The referee has provided some input that will clearly improve the manuscript.

**Major comments**

1. The paper uses too many vague and intuitive descriptions, such as "good" and "poorly". Even though it is a measurement report, it's still a scientific contribution to the community. Therefore, more scientific and quantitative descriptions are required.

**Response:** We agree that the manuscript could use less qualifying language. We have carefully reviewed our descriptions to better quantify the comparisons between different results. More precisely:

- To quantify the agreement between the droplet freezing assays INSEKT and μL-NIPI, the concordance correlation coefficient (CCC) is calculated for each comparison following the method developed by Lin (1989). These values are added to Fig. 15 and will be discussed in the main text:

"In Fig. 15, the INSEKT and μL-NIPI methods are directly compared. To quantify the agreement between the two methods, the concordance correlation coefficient (CCC) is calculated following the method developed by Lin (1989). Reasonable agreement is observed for the full measurement set, with a CCC of 0.81 (Fig.15a). The data obtained during the day (Fig. 15b) shows the best agreement (CCC=0.96), while the data obtained during the night (Fig. 15d) shows less agreement (CCC=0.53). The main deviation in the agreement between the two methods is shown in Fig. 15c (CCC=0.26), which is expected due to a shorter temporal overlap in the sample collection for these two filters. Indeed, the filter for μL-NIPI was collected from 12:20 to 17:00 (UTC+2), while the filter for INSEKT was collected from 16:30 to 20:00 (UTC+2), representing only 30 minutes of overlap between the

measurements. Such short temporal overlap, together with the variations in the aerosol number concentration, could explain the deviation in the INP concentrations measured by the two methods."

- To quantify the agreement between the parameterizations and the measured INP concentrations presented in Fig. 15, the percentage of the data points falling in each parameterizations shaded region will be calculated and the values will be discussed in the main text:

"However, the Schneider et al. (2021) parameterization overestimates the INP concentrations measured at colder temperatures by approximately one order of magnitude, and only 19% of the data points fall within its shaded region (Fig. 15). The Tobo et al. (2013) parameterization, on the other hand, is able to reproduce more of the ambient data, with 35% of the data points within its shaded region. Moreover, although it fails to predict the lowest concentrations obtained from the offline methods, its trend agrees best with both the online and offline INP measurements. Conversely, the DeMott et al. (2010) parameterization only reproduces 3% of the data points and does not capture the ice nucleation behavior observed by the droplet freezing assays at warmer temperatures."

2. There are too many figures than actually needed with repeated information. Some of them are barely mentioned in the text. Please carefully reconsider and prepare the figures.

**Response:** We thank the reviewer for the constructive suggestions for the figures. We agree that some figures share repeated information and we have therefore combined some of them in a finalized manuscript. The details of what we foresee in the finalized manuscript are given under the minor comments on figures, and updated figures are presented at the end of this document.

**Minor comments**

Title: Please explicitly include "instrument inter-comparison" in the title and clarify the INP measurement condition, i.e., mixed-phase cloud formation condition, here and in abstract, and delete the full stop at the end.

**Response:** The title will be modified to include the instrument inter-comparison and to remove the full stop: "Measurement report: Introduction to the HyICE-2018 campaign for measurements of ice nucleating particles and instrument inter-comparison in the Hyytiälä boreal forest".

We will not include information concerning the INP measurement condition in the title in order to keep the title reasonably short. However, such information is added to the abstract as follows:

"In this study, we investigate the INP emission potential from high latitude boreal forests in the mixed-phase cloud regime."

P2L1: Even though not all INP instruments were run simultaneously, it would still be helpful to list the INP instrumentation engaged in this study here instead of using "additional instrumentation for measuring INPs" to give the readers more information on what type of INP instruments were used. Besides, please give the full name of SMEAR according to ACP requirement.

**Response:** We propose to modify the abstract as suggested:

"The campaign utilized the infrastructure of the Station for Measuring Ecosystem-Atmosphere Relations (SMEAR) II, with additional INP instruments, including the Portable Ice Nucleation Chambers I and II (PINC and PINCii), the Spectrometer for Ice Nuclei (SPIN), the Portable Ice Nucleation Experiment (PINE), the Ice Nucleation SpEctrometer of the Karlsruhe Institute of

Technology (INSEKT) and the microlitre Nucleation by Immersed Particle Instrument (µL-NIPI), used to quantify the INP concentrations and sources in the boreal environment."

P2L14: It would be more logical to add a statement about ice formation pathways in the atmosphere (homogeneous and heterogeneous freezing) before introducing INP and heterogeneous ice nucleation.

**Response:** This study focuses on heterogeneous ice nucleation and INP measurements, thus we believe it is reasonable to focus the introduction on these processes without mentioning homogeneous freezing.

P3L14-L15: Please specify the experiment condition (temperature and supersaturation with respect to water) of the organic IN references. Are they relevant to the experiment condition of this study?

**Response**: To keep the text simple, we did not specify the experimental conditions of the references, but we will modify the text to be more specific: "studies have shown that SOA can nucleate ice under cirrus conditions (Wilson et al., 2012; Wagner et al., 2017; Wolf et al., 2020)..."

We acknowledge that the references presented here focus on cirrus conditions, while our measurements focus on mixed-phase cloud conditions. However, we wish to state the current knowledge concerning ice-active SOA, and we do not make any direct comparison to our own study.

P4L4: Please define a.s.l.

**Response:** The acronym will be replaced by "above sea level" as it is used only once. In addition, the acronym "a.g.l" will be replaced by "above ground level" for consistency.

P4L8: Please rephrase the sentence "Multiple towers include…" to "There are several towers…, including …".

**Response:** We will modify the text as suggested.

P5L11-L12: After heating the APS inlet, what did the authors do with the water vapor in the sampling flow? Will the water vapor condense inside the sampling line?

**Response:** It is actually the sampling line that is heated, and not the inlet. The text will be corrected accordingly. Moreover, since the sampling line is heated and the APS itself has a temperature of 35-40°C, moisture does not condense and passes through the instrument without influencing measurements.

P5L14: Please change "operational" to "operating". Also change it on P9L15.

**Response:** We will modify the text as suggested.

P5L16: Please use consistent time format throughout the paper.

**Response:** The time format on P5L16 will be changed to "08:00-20:00, UTC+2" for consistency.

P5L17-L20: This sentence is irrelevant, please remove it.

**Response:** This sentence will be moved to the conclusions where it is more relevant.

P6L1: What makes the sampling "augmented"?

**Response:** We will rename section 2.2 to "Additional INP measurements for the HyICE-2018 campaign" to improve clarity.

P6L12: Please rephrase to "offline IN activity/concentration measurements of particles collected on filters". The original sentence reads as if the authors characterized INPs using the two DFA instruments, which is somehow beyond the instruments' capability.

**Response:** We agree with the reviewer that the second part of the sentence is misleading. We will rephrase it to "two droplet freezing assays [...] were used for offline measurements of INP concentrations collected onto filter samples".

P6L15: Please define PINC, PINCii, and SPIN.

**Response:** We will add one sentence (P6L15) to define the acronyms: "These instruments include the Portable Ice Nucleation Chamber (PINC), the Portable Ice Nucleation Chamber II (PINCii) and the Spectrometer for Ice Nuclei (SPIN)." The titles of the subsections 2.2.1. I, II and III are maintained in order to maintain clarity.

P7L5: Please add a comma before "some particles will…".

**Response**: We will add a comma as suggested.

P7L9: I suggest to replace "frozen and unfrozen particles" to "ice crystal and water droplets".

**Response**: We will modify the text as suggested.

P8L5-L6: Please rephrase "sampling intervals of 5 to minutes…". Did the three CFDCs use different background sampling time?

**Response**: For clarity we will rephrase as follows: "sampling intervals varied between 5 and 20 minutes depending on the instrument, and were separated by background measurements of clean, filtered air".

The three CFDCs indeed used different sampling/background times, as stated P21L15-L18 for the inter-comparison: "The SPIN data is generated from the difference between 15 min sampling averages and 5 min interpolated background concentrations [...] The PINC and PINCii data is processed in an analogous manner to the SPIN data, but with sampling windows of 20 min and 15 min, respectively, and background windows of 10 and 15 minutes, respectively."

P8L11-L12: How were the concentrations during background and measurement sampling window determined? Were they converted from OPC counts and averaged over time? If so, pleas indicate the deduction and average before "concentrations".

**Response**: Yes, the INP concentrations were calculated by converting the OPC counts and averaging over time. We will clarify this by adding text: "Ice crystal concentrations were obtained from the OPC counts and averaged over time. Concentrations obtained during background measurements were then subtracted from the concentrations measured during each sampling window to compute the measured INP concentrations [...]"

P8L24-L25: What's the influence of such a large inlet flow rate compared to the small sampling flow rate? Is it possible that the vacuum pump driving inlet flow sucks from the instruments?

**Response**: The high flow rate through the inlet was simply used as a carrier flow, and it preferentially pulls the air through the inlet in the roof of the cottage since it is the path of least resistance for the vacuum pump. The carrier flow does not influence the flow rates going to each individual instrument since they all have their own dedicated vacuum pumps. We also measured the flow rates to each

instrument while connected to the system to check that the system was functioning properly. We will modify the text P8L24-25 to clarify:

"The inlet was heated to 25-30°C to evaporate droplets and ice crystals and had a carrier flow rate of 250 L min$^{-1}$. Individual instruments then sampled from manifolds on this inlet using their own external pumps."

P8L26: What's the RHw range of the sampling stream at the dryer outlet?

Response: The relative humidity of the sample was kept below 30% (Paramonov et al., 2020). We will add this information P8L26: "A molecular sieve dryer was installed to keep the relative humidity of the sample below 30%".

P9L2: What is the second enrichment factor and how was it determined? Was it for the whole particle size distribution spectrum or for each size?

Response: "Second enrichment factor" was used to differentiate from the PFPC size-dependent enrichment factor. We agree that the phrasing is imprecise and we will modify the text to improve clarity:

"The PFPC concentrates aerosol particles with a certain size-dependent enrichment factor where larger particles are concentrated more efficiently than smaller ones. The size-dependent enrichment factor is determined by measuring the particle size distributions before and after the PFPC. The enrichment factor was estimated as 25 ± 6 for ambient particles of diameters between 0.4 and 2.5 μm when the PFPC was operated at sea level in the vertical configuration (Gute et al., 2019). During the HyICE-2018 campaign, a second enrichment factor was determined before each ice nucleation experiment by calculating the ratio between a concentrated INP measurement point and an ambient measurement point bypassing the PFPC. Ambient INP concentrations were then back-calculated by multiplying the concentrated INP concentrations by this second enrichment factor (Paramonov et al., 2020)."

P9L16: Why is there a 1 °C lamina difference between PINC and PINCii to produce comparable results? How many thermocouples were used to determine PINC and PINCii wall temperature, respectively?

Response: The objective was to operate PINCii with the same parameters as PINC, but later the data analysis showed that PINCii was in fact measuring at 1°C colder. We will remove the end of the sentence on P9L16 to avoid misunderstandings.

Concerning the number of thermocouples used to determine the wall temperature, PINC has 10 thermocouples (4 on each wall of the main chamber and 1 on each wall of the evaporation section) as explained in Chou (2011), while PINCii has 56 thermocouples (7 on each wall of the evaporation section and 21 on each wall of the main chamber) as explained in Castarède et al. (2021, *in preparation*).

P9L17: What is the RHw when the sampling air enter PINCii? Is it possible that the water vapor in the sampling stream condense and affect the measurement and background results at PINC inlet?

Response: We did not measure the RHw at the inlet of PINCii. PINCii was sampling from the main inlet, which was heated to 25-30 °C to evaporate droplets and ice crystals. Moreover, we used a recirculating sheath flow which was dried using a molecular sieve drier before entering the chamber. We monitored the background before and after each ice nucleation measurement and did not detect any background issues related to condensation.

P10L6: If the SPIN design in this study is the same as Garimella et al. 2016, then the statement on P7L4 "drawn through the center of the chamber" should be revised, since SPIN lamina does not lie along the chamber centerline.

Response: We agree with the referee's comment and will remove "center of the" to avoid misunderstandings.

P10L11: Did the authors characterize the VI concentrator? What's the size-resolved enrichment factor? Please add references here. Please also add "(more details about the VI concentrator are provided below)" here.

Response: In an updated manuscript, a figure will be added to the appendix to present the measured magnification factor of the VI concentrator used with SPIN (see also the figures at the end of this document). The corresponding references will also be added as suggested.

P11L2: What is the exact concentration for particles larger than 1.3 μm based on APS data?

Response: The daily average concentration will be added to the text: "...the APS data showed very low number concentrations for particles with diameters larger than 1.3 μm, with a daily average of 0.26 cm$^{-3}$."

P11L3: Why are these large particles more pronounced? Please add references associated with the aerosol types presented in this study.

Response: Several studies have reported dependency between particle surface area and their ice nucleating ability, and it is generally agreed that larger particles make better INPs due to the probability of increased active site occurrence (Archuleta et al., 2005; Augustin et al., 2013; Mason et al., 2016; Hartmann et al., 2016; Reicher et al., 2019). We will update the text to improve clarity:

"Because of such over-representation of larger super-micron particles, and since larger particles with larger surface areas have an increased probability to host active sites, making them better INPs (Mason et al., 2016), it can be expected that the role of such particles as INPs is more pronounced in the SPIN observations."

P11L3-L5: Did the authors measure particle size distribution before and after the VI concentrator with APS?

Response: See our response to the comment on P10L11 for the VI concentrator characterization.

P11L10: RHw = 110% seems to be above the water droplet survival line at -36 °C and -32 °C.

Response: Indeed, compared to other CFDCs, the evaporation section of the SPIN is less efficient due to a shorter residence time. However, it has been experimentally observed that when SPIN is used at high supersaturation above the droplet breakthrough line, the maximum size of liquid droplets is approximately 4.8 μm, with a tail approaching 6 μm (Korhonen et al., 2020). The issues with co-existence of droplets and ice crystals was thus resolved by setting the size separation threshold to 6 μm, where no misinterpretation of droplets for ice crystals is expected. Note that with such high supersaturation, ice crystals typically grow to 8-20 μm with the SPIN settings used during the campaign.

P11L11-L13: The particle concentration varies greatly from time to time in this study according to Fig. 7. It is common practice to fix CFDC lamina condition during field studies because it usually takes

several minutes to change CFDC lamina Si and temperature and re-establish equilibrium. Would the frequent alternation of both temperature and saturation in this study introduce more uncertainty? Are these results obtained with changing ambient condition and lamina condition at the same time comparable with each other? These descriptions might be irrelevant to the focus of this paper, i.e. inter-comparison of different instrumentation. So the author might just remove them.

**Response**: We agree with the referee that frequent changes of CFDC lamina conditions might introduce more uncertainty and would require a detailed data analysis. We do not discuss such limitations here as we are not using the data obtained from the described scans. However, such limitations should/would be discussed in a separate publication including more of the SPIN measurements during HyICE-2018.

Since one objective of this paper is to introduce the HyICE-2018 campaign and the various setups used, we prefer to keep the text as it describes how SPIN was running during the campaign.

P11L19: Please format the reference.

**Response**: We will correct the reference format as suggested.

P11L23-L25: Will this air pumping process through OPC change aerosol population, and therefore INP concentration, in the chamber? Do the authors imply that the aerosols inside PINE chamber distribute homogeneously under changing pressure?

**Response**: The pressure reduction during the pumping process causes a decrease of the aerosol and INP number concentration per air volume in the PINE chamber, which is proportional to the absolute pressure reduction. This reduction is considered when calculating the INP number concentration per air volume at standard conditions. We indeed assume that the PINE chamber volume is always well-mixed for aerosol particles and INPs. The text will be modified to include this information: "[...] leading to a decreasing pressure and a decreasing, but well-mixed, particle number concentration within the chamber."

P12L5: Please clarify the particle transmission efficiency here and in the caption of Fig. 6 are for the sampling system upstream PINE rather than the PINE chamber itself to avoid misunderstanding.

**Response**: We will modify the sentence P12L15 as suggested to improve clarity: "Particle transmission efficiency [...] was investigated by measuring particle concentrations upstream of the PINE chamber inlet and in the ambient air using an OPC (MetOne model GT 526S)"

The Fig.6 caption will also be modified: "Transmission efficiency as a function of particle size for the sampling system upstream PINE."

P12L15-P13L9: Reads like introduction and adds little to the topic. Also between P13L9 and L10, do the authors mean one complete temperature spectrum in one test run?

**Response**: This paragraph introduces the droplet freezing assay techniques, similar to the brief paragraph on the CFDC's working principles. We will shorten and rephrase the text to improve clarity.

In P13L9-L10, we meant that droplet freezing assays provide INP temperature spectra, i.e. INP concentration as a function of temperature, as opposed to online chambers which provide INP concentration at a fixed temperature. We will remove this sentence to shorten the paragraph.

P13L19-L20: Will the cleaning process change filter IN activity?

**Response**: The cleaning process removes particles from the filter surfaces, and thus eliminates foreign INP sources (e.g. from packaging). Pre-cleaning of filters used for aerosol sampling and subsequent INP analysis is a common procedure (Barry et al., 2020). Moreover, tests have been done with handling blanks (pre-cleaned filters which were collected without flowing air through the membranes), and the washing water from the handling blanks usually shows the same INP concentration as filtered nanopure water.

P13L30: How were the collected aerosols washed off and suspended? Mechanically or by supersonic? Please elaborate.

**Response**: The text will be updated to include the missing details and improve clarity: "After sampling, the filters were suspended into 8 mL of nanopure water that had been passed through a 0.1 µm syringe filter. The sample solution was then spun on a rotator for approximately 20 minutes in order to wash the collected aerosol particles off the filter."

P13L31: Please define "PCR".

**Response**: We will define the acronym as suggested.

P13L35: Please add references for the camera and software if it were developed by KIT. Otherwise, please clarify the production company.

**Response**: We will add the requested information: "Brightness changes of the small sample volumes, which correspond to freezing events, were detected using a camera (EO-23122, Edmund Optics Monochrome Camera) and a custom-made LabVIEW program for image acquisition and analysis."

P14L6-L7: It would be better to use "100 ℃ water bath" here. Please also elaborate what container was used to hold the suspensions in the boiling water.

**Response**: The text will be modified as follows: "For the heat treatment tests, a polypropylene test tube (CELLSTAR, Greiner Bio-One) filled with 2 mL of the aerosol suspension was placed in a 100 ℃ water bath for about 20 minutes."

P14L22: Please elaborate.

**Response**: The method was not elaborated here because it was explained in the previous section. We will modify the text to make it clear that the same heat test technique is used for both instruments: "Heat tests were performed using the same technique as described for INSEKT, except the samples were heated for 30 minutes instead of 20 minutes (Hill et al., 2016; O'Sullivan et al., 2018)."

P14L30: Please remove "and INPs". These instruments can only provide vertical profile of ambient aerosols.

**Response**: We did use airplane flights to collect filters at different altitudes and later analyzed their INP content, which gave us some information on the vertical distribution of INPs, as explained in the section 2.3.2 Vertical Profiling.

P15L13: What's the relationship between soot and black carbon?

**Response**: Thank you for spotting this repetition. We will remove "black carbon".

P15L21-L22: What's the transition efficiency of the total aerosol inlet? Will the PM10 inlet prevent large bio-aerosol from entering WIBS? Please elaborate more here.

**Response**: The transmission efficiency of the total aerosol inlet was characterized for PINE measurements as presented in Fig. 6 (moved to appendix). According to this characterization, the WIBS was effectively sampling $PM_5$ aerosols. When the WIBS was moved to the aerosol cottage, it sampled through a PM10 inlet, so particles larger than 10 μm were indeed removed. We will add one sentence to clarify: "Thus the WIBS data reported here is for particles between 0.5 and 5 or 10 μm, depending on the instrument location."

P15L25-L27: What sampler did the authors use? Please add references.

**Response**: We will add the reference for the filter holder used for sampling (P15L26). We are using this sampling protocol for the first time, and for this reason no other reference has been specified.

P15L27-L30: Please add references for the cultivation process.

**Response**: The microbiological cultivation method and the Luria cultivation media are very traditional, broadly used methods and thus commonly presented without references. We will modify the sentence to improve the clarity by adding "or until visible colonies appeared" (P15L29).

P16L12: The parenthesis should appear after "sensors".

**Response**: The parenthesis will be moved as suggested and details will be added for the temperature and relative humidity sensors: "Additional sensors (Rotronic HygroClip-S, PT1-100 temperature sensor and Li-Cor Li-840) measured relative humidity, temperature and $CO_2$ and $H_2O$ concentrations during the flights."

P17L10: Please specify the snow depth and temperature range during winter instead of the vague description "deep snow cover and cold temperature".

**Response**: The average snow depth and the temperature range will be specified as: "During winter, the campaign was characterized by deep snow cover (60 cm in average) and cold temperatures (between -17 and 0 ℃)..."

P17L17: See the comments on these figures.

P21L14: Is there a reason why does PINE exhibit such frequent high variability? Is it valid to average INP concentration measured at different time (~30 min in this case)? Please elaborate.

**Response**: Part of the variability may be natural, but another possibility here is that the low INP concentrations measured on March 22nd are at or near the PINE detection limits, thus where the Poison uncertainties are very large. By averaging we reduce the statistical uncertainty. We will modify the text to improve clarity:

"The PINE data is presented as a 5 point moving average to reduce the uncertainty associated with poor counting statistics in the periods when INP concentrations were close to the detection limit (below $\cong 5$ $L^{-1}$ for a single expansion). The error bars represent 20% uncertainty in absolute INP concentrations (cf. Möhler et al., 2021)"

P21L14: Could the authors change the 20% uncertainty to ±1 standard deviation for PINE results to be consistent with the rest of the text?

**Response**: Here a 20% uncertainty is used for the PINE data to be consistent with the Möhler et al. (2021) PINE technical paper. Moreover, we present the PINE data as a 5-point moving average, which means the statistics for a standard deviation are poor.

P21L19: The trends of INP concentration for PINE and PINC during 18:00-20:00, Mar. 22, and around 11:30, Mar. 28 are opposite. Please explain why does this happen.

**Response**: This comment was very helpful, and we have noted a mistake in the data analysis for March 22 and March 28, where the time of the PINE chamber was UTC+1 and not UTC+2. The data has been corrected and the updated figure is attached at the end of this document. The opposing trends are most likely due to the fact that PINE was measuring with a much higher frequency and was therefore able to capture changes in INP concentrations that were not visible in the PINC data since it gives one data point every 30 minutes. We will modify the text to include such information:

"[...] overall there is good agreement in the INP concentrations measured by the PINE and PINC chambers. The INP concentrations are within the same order of magnitude and generally follow the same trend. On March 22 between 17:00 and 19:00 (UTC+2) and on March 28 around 11:30 (UTC+2), the trends in INP concentrations are however opposite. This might be due to the fact that PINE was measuring with a much higher frequency than PINC and was therefore able to capture changes in INP concentrations that were not visible in the 30-minute PINC data."

P21L20: By a factor of what is the SPIN measured INP concentration lower than that measured by PINE and PINC?

**Response**: We will include a statement to add this information:

"In comparison to the PINC and the PINE chambers, the SPIN chamber tends to measure INP number concentrations lower by a factor of ~10, despite the use of a concentrator."

P22L8-L10: The trends between 12:00-13:00, Apr. 28 are not "very good" and "similar" even the authors attribute the deviations to different experiment temperature choice. Besides, is there a particular reason for the different Tl and T choice?

**Response**: We agree with the reviewer, and we will modify the text to mention the opposite trends between 12:00 and 13:00 on April 28:

"The main deviation is observed on April 28 between 12:00 and 13:00 (UTC+2), where both chambers show opposite trends in the INP concentration. Although this deviation cannot be explained at this time, it is short-lived and only represents a few data points."

There is no particular reason for the difference in temperature Tl and T. We selected the data where PINE and PINCii were running at temperatures as close to one another as possible, although there is still a 3°C difference.

P23L5: Why is the temperature information repeated again and again? Is there any finding in relevance to the daily and long-term temperature variation?

**Response**: The temperature was initially added to provide a meteorological context to the figure after Schneider et al. (2021) showed that long-term INP time series correlated well with ground-level ambient air temperature. However, since the data we present here is limited to several hours, we agree that the

temperature information might not be relevant. The temperature information will be removed from Fig. 11 and the sentence P23L5 will be removed.

P23L25-L26: Does that mean ground-level measurements tend to over-present realistic INP concentration in tropopause?

**Response**: We agree that the sentence is misleading, and we will rephrase to avoid misunderstandings:

"On the other hand, the DeMott et al. (2010) parameterization uses data from nine different campaigns conducted in various environments, sometimes well away from aerosol sources (e.g., in the Arctic), which might explain why it tends to underestimate the INP concentrations measured in the Hyytiälä boreal forest."

P24L2-L3: Given the vast quantities measured in this field campaign, the readers would expect the authors to propose a novel parameterization applicable to boreal forest environment. This would add more to the paper and to the community.

**Response**: A novel parameterization based on the INSEKT measurements during the HyICE-2018 campaign was indeed proposed in Schneider et al., (2021). Another parameterization covering all the INP measurements during HyICE-2018 could be discussed in a future publication of this Special Issue.

P24L7: What measures did the authors take?

**Response**: During the inter-comparison day, two additional filters were collected at the main cottage (in addition to the daily filters collected for both INSEKT and µL-NIPI. We agree that the sentence is misleading, and we will rephrase as follows:

"Although the filters were not collected at the exact same time, efforts were made to coordinate the measurements and two additional filters were collected at the main cottage as a complement to the filters collected on a daily basis."

P24L8: What does "…detects INPs at temperatures up to 5 ℃ warmer…" mean? Does it mean that INSEKT has wider measurement temperature range?

**Response**: We meant to describe the ice onset, and not the temperature detection range. We will modify the text to clarify and to avoid misunderstandings:

"[...] the onset freezing temperatures of the µL-NIPI samples are 5 ℃ lower than for the INSEKT samples [...]"

P24L9-L10: What caused the INP concentration increase with the relatively stable aerosol population and meteorology condition?

**Response**: The reason for the INP concentration increase is unclear, and determining the causality is beyond the scope of this paper. We will however modify the paragraph to improve clarity:

"Although the inter-comparison period was relatively short and filters were collected over 24 hours only, a reasonable temporal agreement is observed and both techniques show lower INP concentrations for the filters that were collected during the night compared to the filters collected exclusively during the day. As reflected in Fig. 11, the aerosol number concentration varies slightly during the day, with a minimum of 500 L$^{-1}$ around 16:00 (UTC+2). Although the variations remain within one order of

magnitude, changes in the aerosol number concentration could explain the differences in the INP concentration measured between the day and night filters."

P25L3-L8: It seems like the dilution process modified the IN active component in the suspension and exhibits a discontinuous drop in Fig. 14. The readers would expect more physicochemical analysis on the aerosol type and abundance in the suspension, rather than a vague and plain statement of "in the range of error bars".

**Response**: We did not have enough suspension to do any physicochemical analysis as suggested by the reviewer and such detail is beyond the scope of this manuscript. The origin of the discontinuity pointed out in Fig. 14 is uncertain but not uncommon. One possible explanation is that, as the sampling time of those filters were comparably short ($\cong$ 4 hours), there might be less INPs in the suspensions, so experimental errors from the preparation of the suspensions (mainly error from pipetting) are amplified. Another possibility is that such discontinuity originates from inhomogeneities in the suspension caused by particle settling in the aliquot from which liquid is removed and diluted in a larger volume, as discussed in Harrison et al. (2018). We will add a statement to the paragraph (P25L3-L8):

"The discontinuous drop observed between the series of data points of both the INSEKT Morning and the INSEKT Afternoon filters occurs at the dilution step and is nonphysical. It might be a consequence of the shorter sampling period used for these two filters (Fig. 13), or of inhomogeneity in the suspension caused by particle settling (Harrison et al., 2018)."

P25L13: Please remove "by".

**Response**: We will modify the text as suggested.

P25L13-L14: What factors caused the large deviation between different sampling time? The authors present diurnal meteorology and particle concentration profile with no analysis and investigation.

**Response**: Please see our answer to Major comment 1 and the comment on P24L9-L10.

P28L5-L6: How was one standard deviation determined for each parameterization? The content in Fig. 16 caption should be elaborated more in the main text rather than not mentioned at all.

**Response**: We will modify the text to include the details mentioned in the caption:

"The selected INP parameterizations are also depicted in the figure, where the shaded regions represent the average aerosol number concentration Np (>0.5 μm) ± 1 standard deviation for the DeMott et al. (2010) and Tobo et al. (2013) parameterizations, and the average air temperature $T_{air}$ ± 1 standard deviation for the Schneider et al. (2021) parameterization. For each parameterization, the average and standard deviation were calculated between 08:00 (UTC+2) on March 28 and 08:00 (UTC+2) on March 29, 2018."

P28L8-L14: Please quantify "good agreement" and "poorly" since this is a scientific paper. The authors may also consider tune down the conclusion here since it is drawn based on one single data point for online instruments.

**Response**: We suggest rephrasing the text as follows:

"As expected, in this context, the Schneider et al. (2021) parameterization performs better at warmer temperatures (between -12 and -25 °C), temperatures for which the parameterization was established. However, it overestimates the INP concentrations measured at colder temperatures by approximately

one order of magnitude, and only 19% of the data points fall within its shaded region (Fig. 15). The Tobo et al. (2013) parameterization, on the other hand, is able to reproduce more of the ambient data, with 35% of the data points within its shaded region. Moreover, although it fails to predict the lowest concentrations obtained from the offline methods, its trend agrees best with both the online and offline INP measurements. Conversely, the DeMott et al. (2010) parameterization only reproduces 3% of the data points and does not capture the ice nucleation behavior observed by the droplet freezing assays at warmer temperatures. However, the DeMott et al. (2010) parameterization is based on various measurements conducted in very different environments, sometimes well away from aerosol sources, and thus bias can be expected."

P28L9-L10: It seems like that the Tobo et al. (2013) parameterization overlaps more because it has larger uncertainty based on the description and figure.

Response: Note that the figure will be updated after a mistake was found in the analysis. The Tobo et al. (2013) parameterization presents a wider shaded area because it is more sensitive to changes in the aerosol number concentrations than the DeMott et al. (2010) parameterization, as suggested by the equations of both parameterizations.

P29L15-L16: Again, what does "…tends to detect INPs at temperatures up to 5 °C warmer…" mean?

Response: See response to the previous comment concerning this phrasing.

P29L16-L18: Too many vague descriptions. Please quantify "good".

Response: Please see our response to the Major Comment 1.

P29L20-L23: Again, please tune down the conclusion here, since it is drawn based on very limited data.

Response: We modified the text as follows:

"The measurements from the inter-comparison days were compared to three existing parameterizations. Although the comparison is based on limited data and thus might not be representative of the entire HyICE-2018 campaign, results show that the DeMott et al. (2010) parameterization tends to underestimate the INP concentrations observed by the online INP chambers for temperatures between -29 and -32 °C, and does not capture the ice nucleation behavior observed by the droplet freezing assays at warmer temperatures, between -20 and -12 °C. The Schneider et al. (2021) parameterization shows better agreement with the INP temperature spectra obtained from the droplet freezing assays, but tends to overestimate the INP concentrations measured by the online chambers between -29 and -32 °C. The parameterization does not capture the daily variability observed in the INP concentration, but such result is not surprising since the parameterization was developed to predict seasonal variation of INP concentrations with a time resolution of one to several days. Finally, the parameterization by Tobo et al. (2013) shows the best agreement with the measured INP concentrations from both online and offline INP measurement techniques, which is attributed to the fact that it is focused on biological aerosol…".

P30L1-L11: Reads like introduction.

Response: Since this measurement report is meant as an introduction to the HyICE-2018 campaign and to future publications related to this campaign, we wanted to conclude by discussing the topics that will be or have been studied in separate publications.

**FIGURES**

Fig. 1:

- The legend text size is too small.

**Response**: The font size will be increased.

Fig. 2:

- Would it be cleaner to add a column in Table 1 and include the sampling period? The daytime average temperature information appeared again in Fig. 7 and Fig. 10. Do the authors intend to show the correlation between temperature and what?

**Response**: We believe that the timeline figure is a common and simple way to represent sampling periods of instruments, and that it conveys the information more clearly than listing sampling periods in Table 1. Here the daytime average temperature provides context and shows how the measurements captured the seasonal transition from winter to spring to summer.

Fig. 3:

Panel a:

- Please specify "main chamber" and "evaporation section" in the plot.

**Response**: We will modify the figure as suggested.

- Supersaturation > 1 means RHi > 200%. I assume the authors use Si to present saturation with respect to ice, rather than supersaturation here? If so, the definitions on P3L26 and P7L5 should also be modified.

**Response**: Indeed, we use Si to represent saturation with respect to ice rather than supersaturation. The definition P3L26 and P7L5 will be modified as suggested.

- It would be better to draw background and measurement sampling periods (green and pink) to scale (e.g., 5 min background to 15 min measurement).

**Response**: Background and measurement sampling periods vary between CFDCs. Here, we arbitrarily decided to represent PINCii's sampling periods consisting of 15 min background and 15 min measurements.

Panel c:

- Please align letter "T" with the arrow on the top right corner.

**Response**: The arrow was intended to symbolize "decreasing" temperature. We will remove both symbols to avoid misunderstanding.

Panel d:

- The legend color of Expansion Chamber doesn't match (brighter red) with the plot.

**Response**: The color will be modified as suggested.

Caption:

- L5-L6: Add a dash between "INP" and "temperature".

**Response**: We use the phrase "INP temperature spectra" (without a dash) throughout the paper and prefer not to change it to maintain clarity. This is standard convention based on previous works (for example: Hill et al., 2016; Schneider et al., 2021).

Fig. 4:

- Please annotate the circled cross mark in the scheme. For example, it could be a virtual valve.

**Response**: Figure 4 will be modified to remove the circled cross mark following the original instrumental set-up plan as represented in Paramonov et al., (2020).

Fig. 5:

- Could the MFCs on the right part of this figure function normally in the setup? Where does the MFCs working pressure difference come from? Is there a needle valve to choke the external pump at the Excess out (free air)?

**Response**: Yes, there was a needle valve to adjust the excess out flow and to provide sufficient pressure difference for the MFCs to function normally. The flow inbound to the VI was much lower than the outbound one (5.3 LPM and 19.5 LPM, respectively) so only little choking of the excess out was needed for proper functionality of the both MFCs. Fig. 5 is modified to include such details.

Fig. 6:

- It adds little to the paper. Could it be moved to appendix?

**Response**: We agree with the reviewer, and we move the figure to the appendix.

Fig. 8:

- The colors and legend position for Snow depth and WIBS total are confusing. It would be better to indicate panel (a) and (b) beside the legends.

**Response**: Figure 8 has been modified and combined with Figs. 7 and 9. The updated figure is presented below.

Fig. 7-9:

- Recombination of the same experiment data set. Could it be reduced to two figures? For example, Fig. 9 could be a panel of Fig. 7.

**Response**: The figure 7, 8 and 9 will be combined to improve clarity. Other comments will be applied as suggested.

Fig. 10:

- This figure is not self-explanatory. Missing legends and too much information placed in the caption.

**Response**: We agree with the reviewer that the figure requires a legend, and we will modify the figure accordingly. The caption will also be shortened to improve clarity.

- It adds little to the main text. There is just one statement about this figure betweeen P17L18-P17L20. Therefore, it might better be moved to appendix.

**Response**: We believe that Fig. 10 is an important figure and would rather keep it in the main text. Some description is therefore added to the text:

"In Fig. 10, aerosol characteristics (concentration and NPF occurrence) are represented as a function of wind direction, air temperature and time for two different heights (8.4 and 67.2 m) below and above the forest canopy. Although trends might be expected due to varying source regions (Tunved et al., 2003, 2006), no clear correlation is observed between the aerosol features, the wind direction and the changing seasons. The same conclusion is drawn for the intermediate heights, 16.8 m and 33.6 m (see Fig. A1 & A2). Such observations further motivate the search for other seasonally dependent variables [...]"

Fig. 11:

- Caption L2: Please replace "with PINCii" with "for PINE and PINCii" to avoid misunderstanding.

**Response**: We will modify the caption as suggested.

- Would it be possible to incorporate inlet information listed in Table 2 in this figure? That way the readers can gain more knowledge regarding the difference between heating and inlet cut-off setups.

**Response**: In order to keep the figure clear and simple, we think that it is better not to incorporate the inlet information in the legend. Some of the inlet settings varied from one day to another, and adding this information to the figure might make it unreadable.

Fig. 12:

- Why don't the authors combine this figure with Fig. 11?

**Response**: Figures 11 and 12 will be combined as suggested.

- If the authors select a temperature window of -29 ℃ to -32 ℃ here to cover the experiment temperature selected in this study, it would be more reasonable to draw two separate lines for -29 ℃ and -31℃ for each parameterization, and draw a shading representing ±1 standard deviation using different particle concentration or ambient temperature for the predicted INP concentration.

**Response**: We indeed selected a temperature window to cover the temperatures selected during the chambers inter-comparison, and we believe that our method is reasonable. Moreover, we calculated the parameterizations with the highest time resolution available (10 minutes for DeMott et al. (2010) and Tobo et al. (2013), and 1 minute for Schneider et al. (2021)). Thus we would not be able to calculate the standard deviation without averaging the data and reducing the time resolution, which we would like to avoid.

Fig. 11 and 12:

- Since this paper concentrate on instrument inter-comparison, the readers would expect a plot like Fig. 15.

**Response**: We agree with the reviewer that such figure would be beneficial. We attempted to produce such scatter plots, but for the online instruments the variability in their time resolutions means that even moderately displaced sampling times become difficult to compare using direct correlation. Moreover,

because the chambers' thermodynamic conditions were not exactly the same, such comparisons might be misleading. We believe that the time series presented in Fig. 8, contextualized with the ambient aerosol measurements, is sufficient to show the agreement between the chambers.

Fig. 13:

- It might be more clear and straightforward to convert this figure to a table.

**Response**: We believe that a timeline figure is clearer than listing the sampling periods in a table. Fig. 13 was also combined to Fig. 14 to improve clarity.

Fig. 14:

- The error bar colour needs to be changed for either instrument.

**Response**: We have tried the suggested modification (see below) but feel that it makes the figure less clear. Therefore, we suggest keeping the figure as it was initially:

[Figure]

Figure 14. Left side: original. Right side: with μL-NIPI error bars in black.

- Why is there a drop of INP concentration around -18 °C for both instruments, even though only INSEKT used the diluted suspension?

**Response**: The discontinuous drop observed in the INP temperature spectra measured with INSEKT is discussed in our response to the comment on P25L3-L8. However, there is no drop in the INP temperature spectra measured with μL-NIPI. Could the reviewer further explain their comment with regards to the μL-NIPI data?

Fig. 15:

- It might be better to use different symbols, rather than colours, to denote different measurement time.

**Response**: We are not sure we understand the reviewer's comment. The color denotes the ice nucleation temperature, as illustrated by the colorbar and as written in the caption. Each panel then represents a different measurement time.

Fig. 16:

- Add "on" before "March 28" in the caption.

**Response**: We will modify the text as suggested.

Table 1:

- A list of abbreviations should be given somewhere.

**Response**: The abbreviations will be added in a note below the table, or are directly given in the text.

- The sampling locations for INSEKT and μL-NIPI are confusing. Are the filters used for both instruments from all three locations?

**Response**: Yes indeed, the filters used for both instruments are from all the three locations.

Note that the quality of the figures presented here is largely decreased due to conversion and will be better in the finalized manuscript.

[revised manuscript text omitted]

Brasseur et al. presented the HyICE-2018 campaign, taking place at one site in the Finnish boreal forest from late February to early June 2018. This is an extensive field campaign involving several institutions and many co-authors. First results from this campaign have been published in Paramonov et al. (2020) and Schneider et al. (2021). Brasseur et al. are now presenting all the instruments that have been installed for the campaign, intercomparing INP concentrations using online and offline measurement techniques, and lay the groundwork for further studies conducted as part of the campaign, which are currently manuscripts in preparation. The manuscript reports measurements within the scope of the journal.

We thank the reference for the useful and careful comments. Our responses are given below each comment.

**Major comments**

In general, the INP concentration inter-comparison is based on only a few data points and hours of measurements. The online comparison measurements take place during the day on four different days, and the offline measurements are made within 24 hours. It would be interesting to learn more about how representative these time spans were in terms of the observed variables (such as INP concentration, aerosol loading, new particle formation events, meteorology, etc.) throughout the entire HyICE-18 campaign. Further questions that are relevant for the manuscripts in preparation might be addressed: Can these results be generalized for the entire campaign? Do you expect similar/different results on other randomly selected days during the campaign?

The objective of the inter-comparison was to quantify the agreement between the instruments when they were measuring under similar thermodynamic conditions at the same time. As the reviewer hints, these time windows were too limited to make conclusions concerning the measured INP concentrations and their link(s) to other variables. Moreover, the inter-comparison days were selected in advance, primarily based on logistical concerns and independently of any ambient conditions such as meteorology and aerosol loading. Determining the representability of the inter-comparison days in terms of the observed variables is beyond the scope of this study, and we leave it to manuscripts where full measurement series are presented (e.g., Paramonov et. al. 2020, Schneider et. al. 2021) to illuminate potential links between INP concentrations and other variables. We expect data from any of the instruments to remain consistent with what was measured during the inter-comparison days given that the same thermodynamic conditions would be used.

It would be helpful to add some information about the findings of the four manuscripts in preparation conducted during the campaign (i.e. P5L20, P9L16, P12L3, P15L31) as a mere listing of the topics discussed in the manuscripts is not useful from the reader's perspective without more information.

**Response**: We understand the reviewer's comment. However, as the manuscripts mentioned are still in preparation and the data analysis and writing are ongoing, it is difficult to summarize their main findings here. Moreover, one of the main objectives of this paper is to introduce the HyICE-2018 campaign, and we listed the topics discussed in future publication as mere information. Note that an overview paper is

foreseen in the same Special Issue once all related papers are published, and that it will summarize the findings from all the publications based on the HyICE-2018 campaign.

One conclusion of the manuscript is that "the DeMott et al. (2010) parameterization tends to underestimate the observed INP concentration while the Schneider et al. (2021) parameterization tends to overestimate the INP concentration and does not capture the daily variability [...]". I have some reservations about drawing this conclusion. First, this statement should be elaborated to certain temperature regimes. Also, these parameterisations are not being used here exactly for what they were intended. For example, the Schneider et al. (2021) parameterization is intended to be predicting daily INP concentrations based on daily air temperature, not for capturing the daily variability. In addition, the DeMott et al. (2010) parameterization, for example, has meanwhile been updated (i.e. DeMott et al., 2015), which is constrained above -20 °C only weekly. I would suggest using the parameterizations as they were intended by their authors before drawing a conclusion.

**Response**: We understand the reviewer's comment and we agree that the statement should be elaborated to certain temperature regimes. We will modify as follows (P29L20):

"The measurements from the inter-comparison days were compared to three existing parameterizations. Although the comparison is based on limited data and thus might not be representative of the entire HyICE-2018 campaign, results show that the DeMott et al. (2010) parameterization tends to underestimate the INP concentrations observed by the online INP chambers for temperatures between -29 and -32 °C, and does not capture the ice nucleation behavior observed by the droplet freezing assays at warmer temperatures, between -20 and -12 °C. The Schneider et al. (2021) parameterization shows better agreement with the INP temperature spectra obtained from the droplet freezing assays, but tends to overestimate the INP concentrations measured by the online chambers between -29 and -32 °C. The parameterization does not capture the daily variability observed in the INP concentration, but such result is not surprising since the parameterization was developed to predict seasonal variation of INP concentrations in a time resolution of one to several days. Finally, the parameterization by Tobo et al. (2013) shows the best agreement with the measured INP concentrations from both online and offline INP measurement techniques, which is attributed to the fact that it focuses on biological aerosol…"

The intent of including the parameterizations was to contextualize the measurements with existing parameterizations and not necessarily to conduct a detailed comparison. We will add such information to the main text to improve clarity (P23L6):

"In Fig. 12, we also present three parameterizations that predict INP concentrations for simple comparison purposes and to give context to our ambient measurements."

Concerning the referee's comment about the DeMott et al. (2010, 2015) parameterizations, we believe that the parameterization from 2015 was developed specifically for mineral dust and does not act as an update from the parameterization from 2010, which is more general. Moreover, the DeMott et al. (2010) parameterization is based on data collected at temperatures between −9 and −35 °C, which is the temperature range for which we use it here.

In line with reviewer 1, I think that the paper uses too many vague descriptions (e.g. P1L4-5: "lower", "higher", P3L27: "any", P8L33 "larger", "smaller", Fig. 4: "see text for details" to state only a few) and the figures often contain repeating results (like in: Figs. 2 & 7, Figs. 7 & 8, Figs. 7 & 9, Figs. 11 & 12), which should be avoided by merging them.

**Response**: As suggested, we have merged the figures containing repeating results. We will also modify the text to improve the descriptions, including:

P1L4-5: "lower" will be replaced by "low and mid" while higher will be replaced by "high"

P3L27 "any" will removed

P8L33 "larger", "smaller: see our response concerning the size-dependent enrichment factor.

Fig. 4: "see text for details": see our response concerning Fig. 4.

**Minor comments**

Please note, I have tried to avoid minor comments that have already been made by reviewer 1.

P2L16 What about pore condensation and freezing (e.g. David et al., 2019)?

**Response**: We will mention the pore condensation and freezing as follows:

"Heterogeneous ice nucleation processes may include (i) deposition nucleation [...] and (v) pore condensation and freezing where ice is formed via liquid water condensation in pores (David et al., 2019)."

P2L21-23 Please add references and state what is known concerning sources and properties in the boreal forest environment, even if only little is known.

**Response**: We will modify the text as follows:

"However, measurements in boreal forests are largely underrepresented and little is known concerning the INP sources and properties from this environment. A recent study showed that the boreal forest is an important source of biogenic INPs, and that the seasonal cycle of INP concentrations is linked to the abundance of biogenic aerosol particles (Schneider et al., 2021)."

P5 Table 1. The abbreviations TSP, PM2.5, PM10, PM… are not defined in the manuscript. Also, I would maybe tell the reader in the caption as a side remark that the information about the APS and the SMPS can be found in Table A1.

**Response:** The abbreviations will be added in the table footnotes and we will mention the APS and DMPS as suggested.

P5L19 What do you mean by "INP signals"?

**Response**: We meant INP concentrations. This sentence will be modified and moved to the conclusions following the comments from the Referee 1.

P6L16 Maybe you could leave "with parallel plate design" away as this is described in the next line in more details.

**Response**: We agree with the referee and will modify the text as suggested.

P6 Fig.3 Some items in the schematic are not clearly defined (e.g. in (a) ice crystal, liquid droplet, in (c) frozen droplet). In (b) "Pressure / Temperature" is a little confusion as you use "INP / L" elsewhere. (d) Maybe you could show the actual operating conditions during your campaign instead of the feasible operating conditions.

**Response**: We will modify the figure as suggested.

P7L9 Please state the inlet size cutoff and refer to the table.

**Response**: This is meant as a general description of the CFDCs principle. The details concerning the sampling conditions during HyICE-2018 are given in the next sections.

P8L29 Maybe refer to Table A1 too?

**Response**: The APS used with PINC is not the SMEAR II APS described in Table A1. We added a reference to Fig. 4 in the description of PINC setup: "...1 L min−1 to PINC (Fig. 4 and Paramonov et al., 2020)."

P8L33 Can you please be more precise regarding the size-dependent enrichment factor here?

**Response**: We will modify the text to include more information regarding the enrichment factor:

"It concentrates aerosol particles with a certain size-dependent enrichment factor where larger particles are concentrated more efficiently than smaller ones. The size-dependent enrichment factor is determined by measuring the particle size distributions before and after the PFPC. The enrichment factor was estimated as 25 ± 6 for ambient particles of diameters between 0.4 and 2.5 µm when the PFPC was operated at sea level in the vertical configuration (Gute et al., 2019)."

P9 Fig.4 What do you mean by "its complete setup"? Also, could you please describe in the caption what the dashed lines represent?

**Response**: We will modify the caption to clarify: "Note that PINC and its complete setup (PFPC, CPC, APS, and SMPS) were used from [...] and were replaced by PINCii from [...]. Between May 4 and May 23, the SMPS was used in parallel of PINCii, as represented with the dashed lines (see text for details)."

P11L4 Can you please elaborate shortly why there was no correction method for the SPIN measurements?

**Response**: The correction method for the SPIN measurements is temperature dependent and the correction factor was not determined for the SPIN measurements during HyICE-2018. As mentioned P22L3-7, such correction factor were discussed in Korhonen et al. (2020) where the SPIN chamber was used for laboratory experiments, and a low bias of ~3 was found at -31°C. However, we cannot know for certain what the correction factor would be for the HyICE-2018 campaign.

Section 2.2.3 Did you do background measurements (e.g. Polen et al., 2018) with INSEKT and uL-NIPI during the campaign? If so, please shortly describe the results.

**Response**: Both handling blank and pure water background measurements were done with INSEKT and µL-NIPI during the campaign. For INSEKT, the INP concentrations reported here were obtained by subtracting the INP concentrations derived from the handling blank filters and the background

measurements. For μL-NIPI, only the data that is statistically above the baseline determined by the handling blanks is shown. We will add a statement P14L5:

"Handling blank filters, which were collected without ambient air flowing through the membranes, were used during the campaign to verify that contamination was not an issue in the absence of ambient particles. Moreover, several control freezing spectra were done using the nanopure water to ensure that the background freezing due to impurities in the water remained low relative to the number of INPs in the water after aerosol sampling. The INP concentrations reported here were obtained by subtracting the INP concentrations derived from both the handling blank filters and the freezing background measurements."

And P14L22:

"Note that, as for INSEKT, handling blank and water background measurements were done to determine the baseline of the results, and only the data that is statistically above the baseline is reported."

P16L14 Please shortly elaborate how boundary layer depth was estimated. What was the mean boundary layer depth during the flights?

Response: We will modify the text to include these details: "The boundary layer depth was estimated during the flights using real-time measurements of particle concentration, relative humidity and temperature, and ranged between 500 and 2500 m (mean = 1300 m, std = 704 m)."

P20 Fig. 10 "right side panel" -> "lower panel"

Response: We will modify the caption as suggested.

P22L7 Can you think of another reason that could maybe explain the rest?

Response: There exists a number of possible sources of error and no clear explanation, other than those already cited, has emerged. For this reason, we report the uncorrected data illustrating the measurement discrepancies between SPIN and the other CFDCs.

P23L12 What temporal resolution was used for the parameterization of Schneider et al. (2021)?

For the figure presenting time series of the online chambers (see new combined figure), we used air temperature measured at 4.2 m with a time resolution of 1 minute.

P24L9 Please rephrase "strong temporal agreement". Be more specific. Do you mean that the concentrations decrease from morning to night samples? I would also be a little careful with the tone of this statement as it is made based on 7 samples collected over 24 hours only. Authors should point this out. Maybe also mention the inconsistent temporal resolution and discuss whether and/or how this might have affected the results.

Response: We will modify the text as follows:

"[...] the INP temperature spectra show substantial overlap between the two techniques. Although the inter-comparison period was relatively short and filters were collected over 24 hours only, a reasonable temporal agreement is observed and both techniques show lower INP concentrations for the filters that were collected during the night compared to the filters collected exclusively during the day."

P25L5 Does this mean that you did not consider the dilution factor in the subsequent calculation of INP concentrations? I would suggest to do so

**Response**: The dilution factor is already considered in the calculation of the INP concentration.

P26 Fig. 14 Please add the appropriate colours for the error bars. Maybe show open symbols for diluted samples.

We attempted the suggested modification for the error bars (as also suggested by referee 1), but feel that it makes the figure less clear (see below). Therefore we prefer keeping the figure as it was initially.

[Figure]

P28 Fig. 16 The parameterizations often have a specific temperature regime for which they are valid. If you extend this temperature range, please make this clearly visible in the figure.

**Response**: We will modify Fig. 16 to make it visible that the temperature range of parameterization from Schneider et al. (2021) was extended:

[Figure]

Figure 11. INP temperature spectra from all the instruments running on March 28. Error bars assigned to online measurement techniques represent the standard deviation of the processed data shown in Fig. 11. The DeMott et al. (2010), Tobo et al. (2013) and Schneider et al. (2021) parameterizations are replotted with the envelopes now indicative of the average N p (>0.5 μm) ±1 standard deviation for DeMott et al. (2010) and Tobo et al. (2013), and the average air temperature Tair±1 standard deviation for the Schneider parameterization. The Schneider et al. (2021) parameterization was extended to -35°C, as shown with the dashed lines.

P32-34 Table A1-3 What do you mean by "Hitu-hut"? Please use consistent wording.

**Response**: "Hitu-hut" is the Finnish word used for "Aerosol cottage". The table has been modified to use consistent wording.